# A quantum register using collective excitations in a Bose–Einstein condensate

Elisha Haber[1,2]*,
Zekai Chen[1,2]†
and Nicholas P. Bigelow[1,2,3]

**1** Department of Physics and Astronomy, University of Rochester, Rochester, New York 14627, USA

**2** Center for Coherence and Quantum Optics, University of Rochester, Rochester, New York 14627, USA

**3** The Institute of Optics, University of Rochester, Rochester, New York 14627, USA

* ehaber@ur.rochester.edu

July 10, 2023

## Abstract

**A qubit made up of an ensemble of atoms is attractive due to its resistance to atom losses, and many proposals to realize such a qubit are based on the Rydberg blockade effect. In this work, we instead consider an experimentally feasible protocol to coherently load a spin-dependent optical lattice from a spatially overlapping Bose–Einstein condensate. Identifying each lattice site as a qubit, with an empty or filled site as the qubit basis, we discuss how high-fidelity single-qubit operations, two-qubit gates between arbitrary pairs of qubits, and nondestructive measurements could be performed. In this setup, the effect of atom losses has been mitigated, the atoms never need to be removed from the ground state manifold, and separate storage and computational bases for the qubits are not required, all of which can be significant sources of decoherence in many other types of atomic qubits.**

## Contents

---

† Current affiliation: Institut für Experimentalphysik und Zentrum für Quantenphysik, Universität Innsbruck, 6020 Innsbruck, Austria

# 1   Introduction

Quantum computers are of interest primarily because they could be used to efficiently solve problems that would be intractable on a classical computer [1]. One promising candidate for a scalable quantum computer researchers have been pursuing is trapped neutral atoms [2–6]. In optical lattices, or arrays of microtraps, atoms are confined to individual traps, and the spatial [7,8] or internal [2–4,9,10] state of the atoms is used to define the qubit. Qubits made up of an ensemble of atoms are more resistant to atom losses and require lower Rabi frequencies than qubits made up of a single atom, but gate operations can be more difficult due to the qubits not being exactly identical to each other, or even the same between realizations [10–14].

Two-qubit gates in lattices can be realized by bringing different qubits close together [3, 15,16], mediating the interaction using a quantum bus (such as an optical cavity [17,18], spin chain [19], or marker atoms [20]), or by using the dipole-dipole interaction between distant Rydberg atoms [4, 17, 21–24]. Qubit gates based on shuttling atoms around or using spin chains are often slower, whereas the Rydberg and optical cavity approaches are faster, but can be prone to a greater variety of errors. Measurements in lattice-based qubits are usually based on resonant absorption or fluorescence in high-resolution optical imaging systems, which allow each qubit to be measured with very high accuracy, but can cause heating [25–28].

Researchers have also considered realizing cold atom qubits using Bose–Einstein condensates (BECs) [29–33], and these proposals are often analogous to different types of superconducting Josephson junction qubits. Although these platforms may be less scalable and suffer from longer gate times compared to lattice-based approaches, it can be easier to read out their states without using resonant light because BECs are mesoscopic objects [34–38]. Additionally, it can be easier to couple BECs (rather than single atoms) to photonic flying qubits [39].

In [40], a BEC in a harmonic trap was used to measure the temperature of atoms confined to an overlapping spin-dependent optical lattice. In [41, 42], atoms hopping between a spin-dependent optical lattice and a harmonic trap were investigated theoretically. In this work, we propose using a BEC in a harmonic trap and atoms in a spatially overlapping optical lattice to realize a quantum register. The goal is to combine the scalability and qubit connectivity of lattice-based platforms with the resistance to atom losses and nondestructive readout available on BEC-based ones.

We begin by discussing a model system and Hamiltonian in section 2, how single and two-qubit gates, and nondestructive measurements, can be realized using this model in section 3, how to achieve such a system experimentally in section 4, and finally several sources of decoherence in section 5.

## 2  The model system

Our goal is to use a system of trapped bosonic atoms as a quantum computer. The Hamiltonian that describes such a system is given by [30, 43],

$$
\begin{aligned}
\hat{H} = \sum_i \int \mathrm{d}\boldsymbol{r}\, \hat{\psi}_i^\dagger(\boldsymbol{r}) \left( -\frac{\hbar^2}{2m}\nabla^2 + V_i(\boldsymbol{r}) + \hbar\omega_i \right) \hat{\psi}_i(\boldsymbol{r}) \\
+ \sum_{i,j} \int \mathrm{d}\boldsymbol{r}\,\mathrm{d}\boldsymbol{r}'\, \hat{\psi}_i^\dagger(\boldsymbol{r}) \hat{\psi}_j^\dagger(\boldsymbol{r}') \tilde{U}_{ij}(\boldsymbol{r},\boldsymbol{r}') \hat{\psi}_i(\boldsymbol{r}) \hat{\psi}_j(\boldsymbol{r}') \\
- \sum_{i\neq j} \int \mathrm{d}\boldsymbol{r} \left( \hbar\tilde{\Omega}_{ij}(\boldsymbol{r},t) \hat{\psi}_i^\dagger(\boldsymbol{r}) \hat{\psi}_j(\boldsymbol{r}) + \mathrm{h.c.} \right),
\end{aligned}
\tag{1}
$$

where the sums are performed over all the internal states, i, of the atoms, $\hat{\psi}_i(\boldsymbol{r})$ is the bosonic field operator for atoms in state i, $m$ is the mass of each atom, $V_i(\boldsymbol{r})$ is the external potential for atoms in i, $\hbar\omega_i$ is the energy of the atom's internal state, $\tilde{U}_{ij}$ is the two-body interaction potential for any pair of atoms, $\tilde{\Omega}_{ij}(\boldsymbol{r},t)$ is the Rabi frequency of any field connecting states i and j, and h.c. denotes the Hermitian conjugate. Note that three-body interactions have been neglected in Eq. (1) [44, 45].

The first term in Eq. (1) is the single-particle Hamiltonian, the second accounts for intra- and interparticle interactions, and the last term allows for transitions between the atomic levels. In writing down Eq. (1), we have assumed that the spacing of the internal energy levels, $\hbar\omega_i$, is sufficiently anharmonic that it is impossible for multiple atoms to collide and all simultaneously change their internal states [40, 46–48].

The two-body interaction potential in Eq. (1) is [46, 47]

$$
\tilde{U}_{ij}(\boldsymbol{r},\boldsymbol{r}') = \frac{4\pi\hbar^2 a_{ij}}{m} \delta(\boldsymbol{r}-\boldsymbol{r}')(1 - \frac{\delta_{ij}}{2}),
\tag{2}
$$

where $a_{ij}$ is the s-wave scattering length for the pair of atoms, $\delta(\boldsymbol{r}-\boldsymbol{r}')$ is the Dirac delta, and $\delta_{ij}$ is the Kronecker delta function.

Eqs. (1) & (2) apply generally to cold bosonic atoms in an external potential. We now specifically consider the potential due to a 3D isotropic harmonic oscillator (HO) and multiple sinusoidal lattices,

$$
V_i(\boldsymbol{r}) = \frac{1}{2} m\omega_{\mathrm{HO}}^2 \boldsymbol{r}^2 + \sum_{l=x,y,z} V_{il} \cos(2\boldsymbol{k}_l \cdot \boldsymbol{r}),
\tag{3}
$$

where $\omega_{\mathrm{HO}}$ is the HO frequency, $V_{il}$ is the state-dependent lattice potential an atom in state i feels due to the lattice along axis l, and $\boldsymbol{k}_l$ is the wavevector of the lattice (which sets the lattice period, $\pi/|\boldsymbol{k}_l|$). Note that all atoms, regardless of their internal state, see the same HO potential. To further simplify our model, and to use the system as a quantum register, we now assume that the lattice potential amplitudes, $V_{il}$, are all either negligibly small, or larger than all other energy scales in the problem.

For atoms in states that see a large lattice potential, we can ignore the (small) HO term in Eq. (3), and if $|\boldsymbol{k}_l|$ and $V_{il}$ are sufficiently large then multiple atoms in the same lattice site will interact very strongly. This means that it will be energetically prohibitive for more than one atom to occupy any of the lattice sites. Additionally, there will be negligible interaction between atoms in adjacent sites because of the large energy barrier, $V_{il}$, between sites.

These assumptions allow us to use the tight binding model to expand the field operators in the Wannier basis [49–53]. We obtain $\hat{\psi}_i(\boldsymbol{r}) = \sum_s w_i(\boldsymbol{r}-\boldsymbol{r}_s)\hat{a}_{si}^\dagger$, where $s$ denotes different

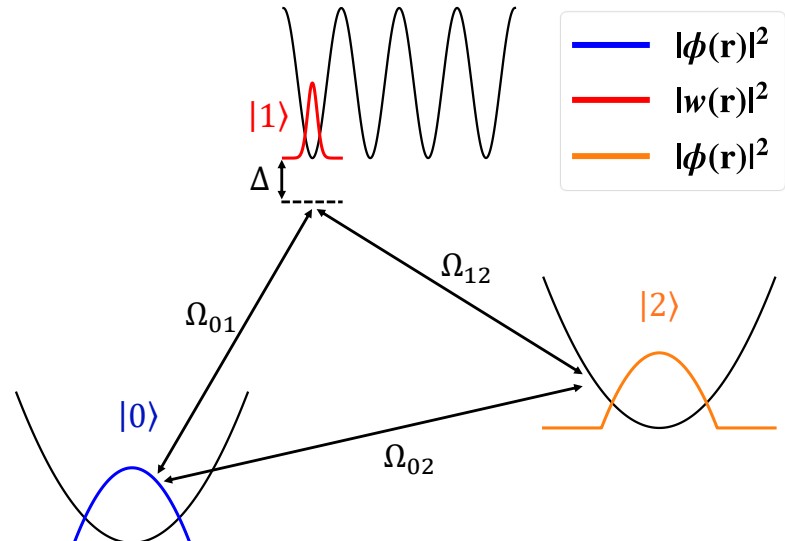

Figure 1: The energy levels, spatial states and fields of the model system. Atoms in internal states $|0\rangle$ and $|2\rangle$ see the HO potential, and their many-body ground state, $\phi(\boldsymbol{r})$, was obtained using the TF approximation. Atoms in $|1\rangle$ see a 3D lattice potential and their spatial states are the Wannier function, $w(\boldsymbol{r})$. The fields that connect the three states together are shown, along with the detuning, $\Delta$, of $\Omega_{01}$ and $\Omega_{12}$ from resonance. The Hamiltonian that describes this system is given by Eq. (5).

lattice sites located at positions $\boldsymbol{r}_s$, $w_i(\boldsymbol{r} - \boldsymbol{r}_s)$ are the Wannier functions, and $\hat{a}_{si}^\dagger$ is the creation operator for an atom in state i and site s. This basis is desirable because each Wannier function is maximally localized to an individual lattice site (see Fig. 1).

For atoms in the internal states where all $V_{il} \ll \hbar\omega_{HO}$ (i.e. for atoms that only see the HO), we assume that our system is initialized with the large majority of these atoms in the many-body ground state of the HO (i.e. that the atoms have condensed into a BEC) [50, 54]. Transfer into and out of this highly populated ground state, unlike transfer to the higher-lying levels, will be enhanced by the $\sqrt{N}$ bosonic creation/annihilation factor, where $N$ is the ground state population. We can therefore, to good approximation, ignore the higher-lying levels and replace the bosonic field operators with one creation/annihilation operator that adds/removes atoms from the many-body ground state of the HO. We obtain $\hat{\psi}_i(\boldsymbol{r}) = \phi_i(\boldsymbol{r})\hat{a}_i^\dagger$, where $\hat{a}_i^\dagger$ creates an atom in the internal state i, and the spatial state $\phi_i(\boldsymbol{r})$. Following [50, 54], we use the Thomas–Fermi (TF) approximation to find the many-body ground state wavefunction, $\phi_i(\boldsymbol{r})$, from Eq. (1) in the mean-field limit (also known as the Gross–Pitaevskii equation). $\phi_i(\boldsymbol{r})$ is plotted in Fig. 1.

By substituting Eqs. (2) and (3) into (1), expanding the field operators as discussed above, and then integrating over all space, we obtain the Bose–Hubbard Hamiltonian [30, 50–53],

$$\hat{H} = \sum_{i,s}\left(\frac{1}{2}U_{is,is}\hat{n}_{is}(\hat{n}_{is}-1) + \epsilon_{is}\hat{n}_{is} + \sum_{i\neq j,r}\left[U_{is,jr}\hat{n}_{is}\hat{n}_{jr} - \hbar\left(\Omega_{is,jr}(t)\hat{a}_{is}^\dagger\hat{a}_{jr} + \text{h.c.}\right)\right]\right), \quad (4)$$

where i and j denote internal states, r and s denote the spatial states of the atoms (given by either the Wannier or the TF wavefunctions), $U_{is,jr}$ is the interaction strength between an atom in state i, s and one in j, r, $\hat{n}_{is} = \hat{a}_{is}^\dagger\hat{a}_{is}$ where $\hat{a}_{is}^\dagger$ is the creation operator for atoms in state i, s, $\epsilon_{is}$ is the total energy of an atom in state i, s, and $\Omega_{is,jr}$ is the Rabi frequency for a transition between states i, s and j, r. Note that for atoms that see the lattice potential, the summations

over r and s are over all the sites of the lattice, while for the atoms in the HO, r and s can only denote a single state (the many-body ground state).

For most of our analysis, it will be sufficient to restrict our attention to three internal states, labeled $i = 0, 1, 2$, and a single lattice site (which we now explicitly restrict to have an occupancy of at most one atom). If atoms in states $|0\rangle$ and $|2\rangle$ only see the HO, and atoms in $|1\rangle$ see both the HO and the lattice, then Eq. (4) simplifies to

$$
\begin{aligned}
\hat{H} = &\frac{1}{2} U_{00} \hat{n}_0 (\hat{n}_0 - 1) + \frac{1}{2} U_{22} \hat{n}_2 (\hat{n}_2 - 1) \\
&+ U_{01} \hat{n}_0 \hat{\sigma}_1 + U_{12} \hat{n}_2 \hat{\sigma}_1 + U_{02} \hat{n}_0 \hat{n}_2 + \hbar \Delta \hat{\sigma}_1 \\
&- \hbar \left( \Omega_{01} \hat{a}_0^\dagger \hat{\sigma}_- + \Omega_{12} \hat{a}_2 \hat{\sigma}_+ + \Omega_{02} \hat{a}_0^\dagger \hat{a}_2 + \text{h.c.} \right),
\end{aligned}
\tag{5}
$$

where $\hat{\sigma}_+$ and $\hat{\sigma}_-$ are the creation and annihilation operators for atoms in state $|1\rangle$, $\hat{\sigma}_1 = \hat{\sigma}_+ \hat{\sigma}_-$ is the number operator for atoms in state $|1\rangle$, and $\Delta$ is the detuning of the fields $\Omega_{01}$ and $\Omega_{12}$ from resonance (see Fig. 1). Note that we assumed $\Omega_{01}$ and $\Omega_{12}$ are detuned the same (small) amount from resonance, and that $\Omega_{02}$ has no detuning. This allowed us to make the rotating-wave approximation to both remove the time-dependence from the Rabi frequencies, and to adjust the energies of the internal states [55, 56].

We note that the interaction energies and Rabi frequencies given in Eq. (5) are related to those in (1) by the Franck–Condon-like factors,

$$
\begin{aligned}
U_{ij} &= \int \mathrm{d}\boldsymbol{r} \, \mathrm{d}\boldsymbol{r}' \psi_i^\dagger(\boldsymbol{r}) \psi_j^\dagger(\boldsymbol{r}') \tilde{U}_{ij}(\boldsymbol{r}, \boldsymbol{r}') \psi_i(\boldsymbol{r}) \psi_j(\boldsymbol{r}'), \\
\Omega_{ij} &= \int \mathrm{d}\boldsymbol{r} \, \psi_i^\dagger(\boldsymbol{r}) \tilde{\Omega}_{ij}(\boldsymbol{r}) \psi_j(\boldsymbol{r}),
\end{aligned}
\tag{6}
$$

where

$$
\psi_i(\boldsymbol{r}) = \begin{cases} \phi(\boldsymbol{r}), & i = 0, 2 \\ w(\boldsymbol{r} - \boldsymbol{r}_0), & i = 1 \end{cases},
$$

and $\boldsymbol{r}_0$ is the location of the lattice site occupied by atoms in $|1\rangle$.

We therefore see that the coupling strengths $\Omega_{01}$ and $\Omega_{12}$ will be reduced from their free-space analogs, $\tilde{\Omega}_{01}(\boldsymbol{r})$ and $\tilde{\Omega}_{12}(\boldsymbol{r})$ (the bare Rabi frequencies), by an amount set by the overlap between the TF wavefunction, $\phi(\boldsymbol{r})$, and the Wannier function, $w(\boldsymbol{r} - \boldsymbol{r}_0)$. Earlier, we assumed that the lattice potentials that are nonzero are much greater than the harmonic oscillator energy ($V_{il} \gg \hbar \omega_{HO}$), which implies that the length scale of the TF wavefunction, $\phi(\boldsymbol{r})$, is much greater than the length scale of the Wannier function, $w(\boldsymbol{r} - \boldsymbol{r}_0)$. This implies that there will be significant spatial overlap between $\phi(\boldsymbol{r})$ and the Wannier functions, $w(\boldsymbol{r} - \boldsymbol{r}_s)$, for many lattice sites, s. In section 4, this spatial overlap will make it possible for us to realize a fully connected register of 1,000 qubits.

We will next show how to use the Hamiltonian given by Eq. (5) to realize a quantum register upon which arbitrary one and two-qubit operations, and nondestructive qubit state readout, can be performed.

# 3  Qubit operations

## 3.1  Single-qubit gates

To implement single-qubit gates, we begin by only turning on the field $\Omega_{01}$, in which case state $|2\rangle$ can be removed from the dynamics. We also restrict our attention to a single lattice site, which we will use as our qubit. The Hilbert space will therefore be spanned by only two Fock states, which we define to be our two qubit states: $|\downarrow\rangle = |N_0, 0, N_2\rangle$ and $|\uparrow\rangle = |N_0 - 1, 1, N_2\rangle$, where the first number in each state is the number of atoms in state $|0\rangle$, the second is the number of atoms in $|1\rangle$, and the third the number of atoms in $|2\rangle$. In matrix form, the Hamiltonian given by Eq. (5) is now

$$H_1 = \begin{matrix} |\downarrow\rangle \\ |\uparrow\rangle \end{matrix} \begin{pmatrix} (U_{00} - U_{01})(N_0 - 1) + (U_{02} - U_{12})N_2 & -\sqrt{N_0}\hbar\Omega_{01}^* \\ -\sqrt{N_0}\hbar\Omega_{01} & \hbar\Delta \end{pmatrix}, \tag{7}$$

and we see that we can in principle create any arbitrary superposition of qubit states by tuning $\Omega_{01}$ and $\Delta$ [1]. This choice of qubit states means that single-qubit gates correspond to coherently taking one atom out of the BEC in the HO and placing it in a particular lattice site, or vice versa. Each single-qubit operation leads to every atom in the BEC being placed in the same superposition of occupying or not occupying that particular lattice site. A second single-qubit operation on a different site will produce an even larger superposition in which every atom has some probability of occupying each of the two lattice sites. In this way, the BEC acts as a reservoir of excitations for the array of qubits. In sections 4 and 5, we discuss several of the advantages and disadvantages of defining the computational basis states in this way. In section 4.2, we also discuss how different sites may be individually addressed in our lattice, and thus how a quantum register with multiple, independently controllable qubits could be realized.

## 3.2  CNOT gate

In order to entangle multiple qubits in the lattice together, we consider using the BEC as a native quantum bus. Using a bus to mediate the interaction between atoms in the lattice is necessary because, in order to derive our model Hamiltonian given by Eq. (5), we assumed that atoms in neighboring sites do not interact. In this section, we show how to realize a CNOT gate, which flips the state of one (target) qubit in the register if and only if another (control) qubit is in the $|\uparrow\rangle$ state. Note that the control and target qubits correspond to different sites in the lattice. We chose the CNOT gate as an example because arbitrary one-qubit gates plus the CNOT gate form a universal gate set [1, 57].

In the first step of the protocol, we entangle the BEC with the control qubit by connecting the BEC states, $|0\rangle$ and $|2\rangle$, via a two-photon virtual transition through the lattice state, $|1\rangle$, using the $\Omega_{01}$ and $\Omega_{12}$ fields (see Fig. 2). This is similar to the two-photon Raman process commonly used in atomic physics [58–60]. Examining Eq. (5), we see that if $\Omega_{01}\sqrt{N_0}, \Omega_{12}\sqrt{N_2} \ll \Delta$ then the energy spectrum will be well-separated into two distinct manifolds. The first manifold contains the Fock states where the control qubit is in the state $|\downarrow\rangle$ (i.e. the lattice site is empty, $\sigma_1 = 0$), and the second manifold the states where the control qubit is in the state $|\uparrow\rangle$ (i.e. the site is filled, $\sigma_1 = 1$).

We may calculate two effective Hamiltonians that each act on one of these manifolds by treating the $\Omega_{01}$ and $\Omega_{12}$ terms as a perturbation in Eq. (5). The result, valid to second order in perturbation theory, (which could also be obtained from a Schrieffer–Wolff transformation [61]) is [56]

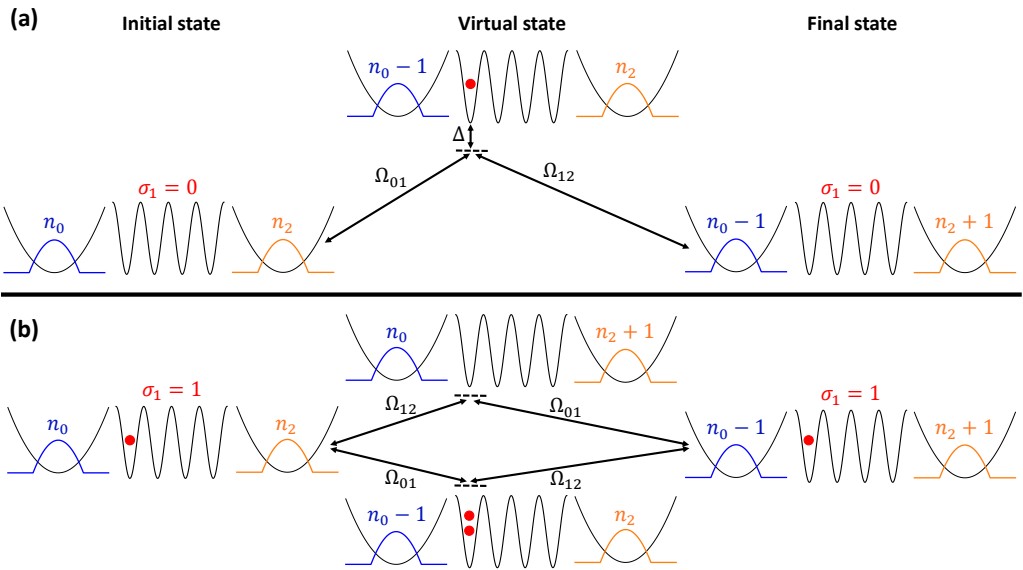

Figure 2: The two-photon process used to couple the BEC state to the state of the control qubit. In (a) the control qubit is in $|\downarrow\rangle$, which means that when the $\Omega_{01}$ and $\Omega_{12}$ fields are applied there is (to first order) only one intermediate state. For large $\Delta$, the system will undergo virtual transitions and there will be significant population flow between the two BEC states. In (b) the control qubit is in $|\uparrow\rangle$ and consequently there are now two paths by which an atom could transfer between the BEC states. For the appropriate choice of $\Delta$, there will be complete destructive interference between these two paths, and there will be no population flow.

$$
\begin{aligned}
\langle n_0, \sigma_1, n_2 | \hat{H}_{\text{eff}}^{\sigma_1} | n_0', \sigma_1, n_2' \rangle = & \, E_{n_0, \sigma_1, n_2} \delta_{n_0 n_0'} \delta_{n_2 n_2'} \\
& + \frac{1}{2} \sum_{s_1 \neq \sigma_1} \sum_{m_0, m_2} \langle n_0, \sigma_1, n_2 | \hat{\Omega} | m_0, s_1, m_2 \rangle \langle m_0, s_1, m_2 | \hat{\Omega} | n_0', \sigma_1, n_2' \rangle \\
& \times \left( \frac{1}{E_{n_0 \sigma_1 n_2} - E_{m_0 s_1 m_2}} + \frac{1}{E_{n_0' \sigma_1 n_2'} - E_{m_0 s_1 m_2}} \right),
\end{aligned} \tag{8}
$$

where

$$
\begin{aligned}
E_{n_0, s_1, n_2} = & \, \langle n_0, s_1, n_2 | \hat{H} | n_0, s_1, n_2 \rangle \\
= & \, \frac{1}{2} U_{00} n_0 (n_0 - 1) + \frac{1}{2} U_{11} s_1 (s_1 - 1) + \frac{1}{2} U_{22} n_2 (n_2 - 1) \\
& + U_{01} n_0 s_1 + U_{12} n_2 s_1 + U_{02} n_0 n_2 + \hbar \Delta s_1, \\
\hat{\Omega} = & -\hbar \left( \Omega_{01} \hat{a}_0^\dagger \hat{\sigma}_- + \Omega_{12} \hat{a}_2 \hat{\sigma}_+ + \text{h.c.} \right),
\end{aligned}
$$

the double summation is over every state outside the $\sigma_1$ manifold ($s_1$ runs over the possible numbers of atoms in state $|1\rangle$ excluding $\sigma_1$, and $m_0$ and $m_2$ run over all the possible numbers of atoms in states $|0\rangle$ and $|2\rangle$, respectively). The effective Hamiltonian, $\hat{H}_{\text{eff}}^{\sigma_1}$, tells us how the populations of $|0\rangle$ and $|2\rangle$ will respond to the $\Omega_{01}$ and $\Omega_{12}$ fields for a fixed qubit state ($\sigma_1 = 0$ or 1). Note that the vast majority of the terms of the double summation in Eq. (8) are zero, since $\hat{\Omega}$ only connects states where a single atom has moved from $|0\rangle$ or $|2\rangle$ and into $|1\rangle$.

When $\sigma_1 = 0$, each state, $|n_0, 0, n_2\rangle$, will be connected to two states by Eq. (8): $|n_0 - 1, 1, n_2\rangle$ and $|n_0, 1, n_2 - 1\rangle$, which will also be connected to $|n_0 - 1, 0, n_2 + 1\rangle$ and $|n_0 + 1, 0, n_2 - 1\rangle$ (see Fig. 2(a)). In the limit of large detuning, we therefore see that population may be able to flow between $|0\rangle$ and $|2\rangle$ via a two-photon process.

We now assume that $n_0, n_2 \gg 1$ during the dynamics (i.e. that the BEC components in $|0\rangle$ and $|2\rangle$ never approach 1 as they exchange population), and that $\Omega_{01} = \Omega_{12} = \Omega$. In this case, the coupling terms connecting $|n_0, 0, n_2\rangle$ to $|n_0 \pm 1, 0, n_2 \mp 1\rangle$, calculated using Eq. (8), will be much larger than the differential AC Stark shifts between these three states (the Stark shift for the state $|n_0, 0, n_2\rangle$ is given by the summation term in in Eq. (8) when $n_0' = n_0$ and $n_2' = n_2$). Using Eq. (8), we find that the transition elements in $\hat{H}_{\text{eff}}^0$ are given by

$$\langle n_0, 0, n_2| \hat{H}_{\text{eff}}^0 |n_0 + 1, 0, n_2 - 1\rangle =$$
$$\frac{\frac{1}{2}\hbar^2\Omega^2\sqrt{(n_0+1)n_2}}{-\hbar\Delta + (N-1)(U_{02} - U_{12}) + n_0(U_{00} - U_{01} - U_{02} + U_{12})}$$
$$+ \frac{\frac{1}{2}\hbar^2\Omega^2\sqrt{(n_0+1)n_2}}{-\hbar\Delta + (N-1)(U_{22} - U_{12}) + n_0(U_{02} - U_{01} - U_{22} + U_{12})},$$

where we substituted $N = n_0 + n_1$. The denominator of each fraction contains three terms, and only the last one, which is proportional to $n_0$, will change for different choices of $n_0$ and $n_2$. For the $^{87}$Rb atoms we consider in section 4, the interaction energies $U_{02}$, $U_{01}$, $U_{22}$ and $U_{12}$ will all be almost identical, and as a result, the denominators will be dominated by the first two terms (and thus be nearly constant).

If the terms proportional to $n_0$ are dropped completely, then we obtain

$$\langle n_0, 0, n_2| \hat{H}_{\text{eff}}^0 |n_0 + 1, 0, n_2 - 1\rangle \approx J_0 \sqrt{(n_0+1)n_2}, \tag{9}$$

where

$$J_0 = \frac{1}{2}\hbar^2\Omega^2\left(\frac{1}{-\hbar\Delta + (N-1)(U_{02} - U_{12})} + \frac{1}{-\hbar\Delta + (N-1)(U_{22} - U_{12})}\right)$$

is not a function of $n_0$ or $n_2$. We can do an identical calculation for the AC Stark shift of each Fock state. After making the same approximations we did above, we find

$$\langle n_0, 0, n_2| \hat{H}_{\text{eff}}^0 |n_0, 0, n_2\rangle - \langle n_0 + 1, 0, n_2 - 1| \hat{H}_{\text{eff}}^0 |n_0 + 1, 0, n_2 - 1\rangle \approx E_{n_0, 0, n_2} - E_{n_0+1, 0, n_2-1} \tag{10}$$

(i.e. the energy differences between states separated by one atom hopping between $|0\rangle$ and $|2\rangle$ are approximately equal to their unperturbed values).

Using the approximations in Eqs. (9) & (10), $\hat{H}_{\text{eff}}^0$ is analogous to the Hamiltonian of the internal Josephson effect/double-well [43, 53, 62, 63]. Under the single-mode approximation (SMA) [64], the entire population may be continuously transferred between the two wells/internal states [65]. The dynamics under Eq. (8) are plotted in Fig. 3. Almost complete population transfer between $|0\rangle$ and $|2\rangle$ occurs in the $\sigma_1 = 0$ case.

In the $\sigma_1 = 1$ case, $|n_0, 1, n_2\rangle$ will be connected to four states by Eq. (8): $|n_0 + 1, 0, n_2\rangle$, $|n_0, 0, n_2 + 1\rangle$, $|n_0 - 1, 2, n_2\rangle$ and $|n_0, 2, n_2 - 1\rangle$. These four states will further be connected to $|n_0 \pm 1, 1, n_1 \mp 1\rangle$ (see Fig. 2(b)), which means that, as when $\sigma_1 = 0$, population may be able to flow between $|0\rangle$ and $|2\rangle$ via a far-detuned two-photon process.

As before, we assume $n_0, n_2 \gg 1$, set $\Omega_{01} = \Omega_{12} = \Omega$, and use Eq. (8) to calculate the transition elements of $\hat{H}_{\text{eff}}^1$, which are given by

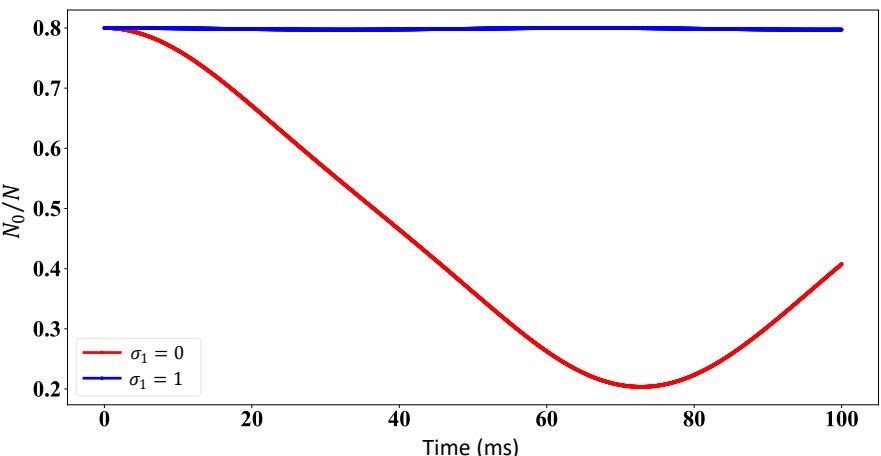

Figure 3: BEC dynamics under the Hamiltonian given by Eq. (8). The dynamics were obtained in the mean-field limit by replacing the atomic creation and annihilation operators in Eq. (8) with complex numbers, and then numerically solving the Heisenberg equations of motion for the population imbalance and relative phase between $|0\rangle$ and $|2\rangle$ [53, 62]. The experimental parameters that were used to generate the plot are given in section 4.2.

$$\langle n_0, 1, n_2 | \hat{H}_{\text{eff}}^1 | n_0 + 1, 1, n_2 - 1 \rangle =$$

$$\frac{\hbar^2 \Omega^2 \sqrt{(n_0 + 1)n_2}}{-\hbar\Delta + (N/2)(U_{00} - U_{01} + U_{02} - U_{12}) + U_{01} - U_{02} - U_{11} + U_{12} - n_0(U_{00} - U_{01} - U_{02} + U_{12})}$$

$$+ \frac{\hbar^2 \Omega^2 \sqrt{(n_0 + 1)n_2}}{-\hbar\Delta + (N/2)(-U_{01} + U_{02} - U_{12} + U_{22}) - U_{11} - U_{22} + 2U_{12} - n_0(U_{02} - U_{01} - U_{22} + U_{12})}$$

$$+ \frac{\frac{1}{2}\hbar^2 \Omega^2 \sqrt{(n_0 + 1)n_2}}{\hbar\Delta + (N/2)(-U_{00} + U_{01} - U_{02} + U_{12}) + n_0(U_{00} - U_{01} - U_{02} + U_{12})}$$

$$+ \frac{\frac{1}{2}\hbar^2 \Omega^2 \sqrt{(n_0 + 1)n_2}}{\hbar\Delta + (N/2)(U_{01} - U_{02} + U_{12} - U_{22}) + U_{01} - U_{02} - U_{12} + U_{22} + n_0(U_{02} - U_{01} - U_{22} + U_{12})}.$$

As in the $\sigma_1 = 0$ case, each term that is proportional to $n_0$ will be negligibly small, allowing us to write

$$\langle n_0, 1, n_2 | \hat{H}_{\text{eff}}^1 | n_0 + 1, 1, n_2 - 1 \rangle \approx J_1 \sqrt{(n_0 + 1)n_2},$$

where

$$J_1 =$$

$$\frac{\hbar^2 \Omega^2}{-\hbar\Delta + (N/2)(U_{00} - U_{01} + U_{02} - U_{12}) + U_{01} - U_{02} - U_{11} + U_{12}}$$

$$+ \frac{\hbar^2 \Omega^2}{-\hbar\Delta + (N/2)(-U_{01} + U_{02} - U_{12} + U_{22}) - U_{11} - U_{22} + 2U_{12}}$$

$$+ \frac{\frac{1}{2}\hbar^2 \Omega^2}{\hbar\Delta + (N/2)(-U_{00} + U_{01} - U_{02} + U_{12})}$$

$$+ \frac{\frac{1}{2}\hbar^2 \Omega^2}{\hbar\Delta + (N/2)(U_{01} - U_{02} + U_{12} - U_{22}) + U_{01} - U_{02} - U_{12} + U_{22}},$$

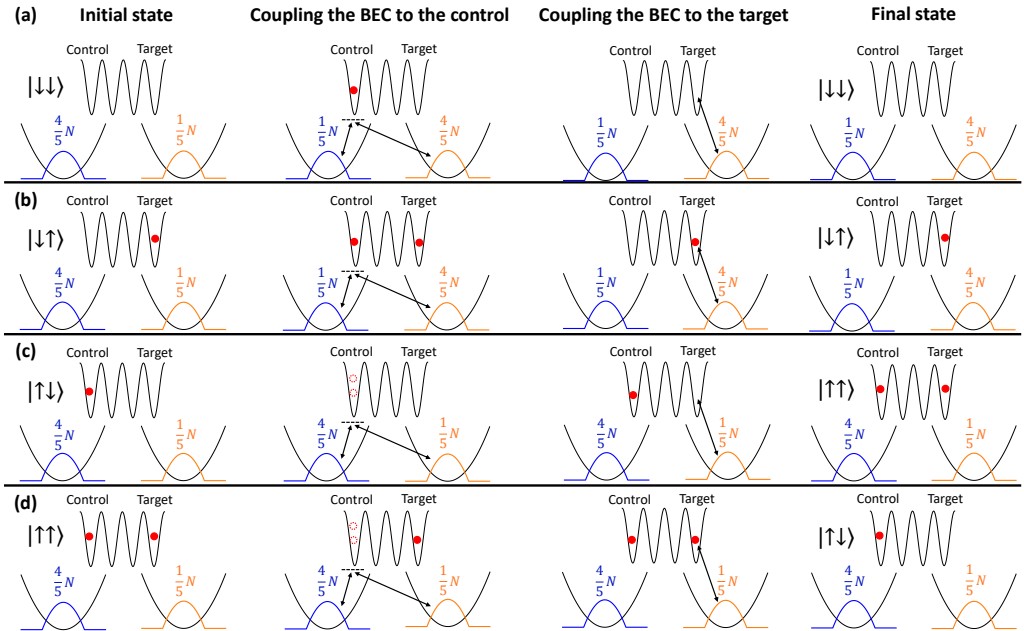

Figure 4: Steps to implement a CNOT gate. The first column shows the possible two-qubit combinations, and the initial BEC populations. The first step of the protocol (second column) is the same process that is shown in Fig. 2. The next step (third column) is to perform a single-qubit operation on the target (see section 3.1). Since $N_0$ is now a function of the state of the control qubit, this effectively couples the control and target qubits together. During this step, the field is left on just long enough so that the target qubit's state is flipped if the control was in $|\uparrow\rangle$.

which is not a function of $n_0$ or $n_2$. The difference between $J_0$ and $J_1$ arises from the interference between the two paths by which an atom may hop between $|0\rangle$ and $|2\rangle$ in the $\sigma_1 = 1$ case (see Fig. 2). For an appropriate choice of $\Delta$, this interference will be destructive and $J_1$ will be close to zero while $J_0$ remains nonzero (see Fig. 3). Thus, the BEC population in $|0\rangle$ and $|2\rangle$ will be a function of the control qubit's state.

Although not explicit in our discussion so far, the above analysis is conditional on the population of states $|0\rangle$ and $|2\rangle$ always being much greater than the population of the higher-lying levels of the HO. This is to avoid significant coupling between state $|1\rangle$ and these higher-lying levels over the timescales of interest. Assuming all of the population initially begins in either $|0\rangle$ or $|2\rangle$, a significant fraction may initially be transferred directly to the other state using the $\Omega_{02}$ field. The two-photon process described above may then be used to entangle the BEC with the control qubit. As discussed in section 4.3, being able to quickly entangle the state of the BEC with a qubit provides a convenient method for nondestructively measuring the state of any qubit in our register.

Finally, we wish to change the state of the target qubit based on the state of the BEC. This can be done using the single-qubit protocol from section 3.1. In Eq. (7) the Rabi frequency for single-qubit operations is proportional to $\sqrt{N_0}$. Since $N_0$ is now a function of the state of the control qubit, performing a single-qubit operation on the target will lead to the state of the target qubit being dependent on the state of the control one.

In order to realize a CNOT gate, we initialize the system with $N_0 = 4N/5$ and $N_2 = N/5$, where $N$ is the total BEC population (see Fig. 4). Using $\Omega_{01}$ and $\Omega_{12}$, we then drive the two-photon transition connecting levels $|0\rangle$ and $|2\rangle$ as shown in Fig. 2. We leave the fields on until $N_0 = N/5$ and $N_2 = 4N/5$ (assuming the control qubit is in $|\downarrow\rangle$). If the control qubit is in $|\uparrow\rangle$,

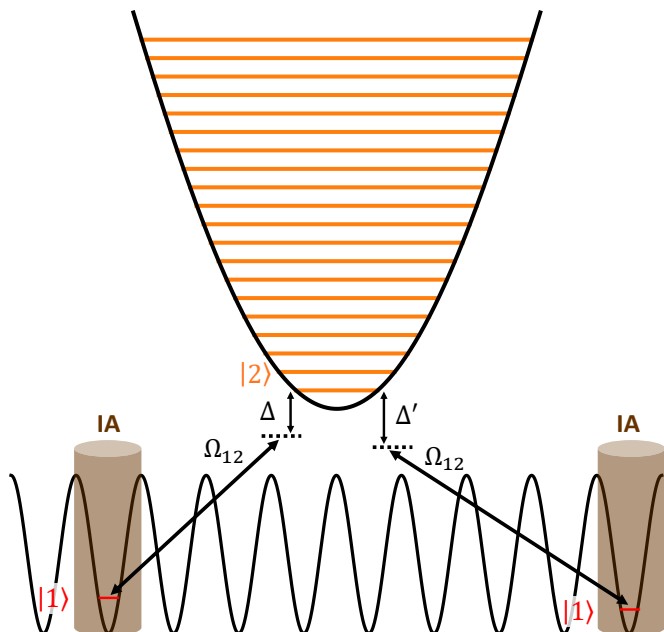

Figure 5: Energy levels and fields involved in realizing a $\sqrt{\text{SWAP}}$ gate. $\Omega_{12}$ connects $|1\rangle$ and $|2\rangle$ with detuning $\Delta$ or $\Delta'$. The IA beams are used to address individual lattice sites. The horizontal line in each lattice site represents the energy an atom in state $|1\rangle$ would have in that site, and the first 20 (nondegenerate) 3D HO levels are shown in the harmonic potential, which are the energy levels available to an atom in state $|2\rangle$.

then there will be no population change.

If the control qubit is in the $|\!\downarrow\rangle$ state, then the BEC population in state $|0\rangle$ will be $N_0 = 4N/5$. Plugging this into Eq. (7), we turn on the $\Omega_{01}$ field just long enough to perform a $2\pi$ pulse on the target qubit (thus leaving its state unchanged). If the control qubit is in the $|\!\uparrow\rangle$ state, however, then $N_0 = N/5$, so we see by Eq. (7) that the Rabi frequency will only be half as large in this case. The result is that the target will experience a $\pi$ pulse if and only if the control is in the $|\!\uparrow\rangle$ state. After we reset the BEC populations (by repeating the operation shown in the second column of Fig. 4), we will have successfully realized a CNOT gate.

## 3.3 $\sqrt{\text{SWAP}}$ gate

Instead of using the BEC to couple qubits that are far apart, as in section 3.2, it is possible to instead use an empty mode of the HO to mediate the interaction (similar to how two qubits placed in an empty cavity can become entangled [66]). Since the BEC is not being used to entangle the qubits this gate, unlike the CNOT gate, could in principle be applied to multiple pairs of qubits in parallel.

By turning on the $\Omega_{12}$ field with a large detuning, $\Delta \gg \Omega_{12}$, and if level $|2\rangle$ is unoccupied, then it becomes possible for atoms to be destroyed in one lattice site and created in another by intermediately occupying the virtual level $|2\rangle$ (see Fig. 5). In this section, we will show how this process may be used to realize a $\sqrt{\text{SWAP}}$ gate which, together with single-qubit operations, forms a universal gate set [66].

To model this process, we must return to Eq. (1), and take into account the spin states $|1\rangle$ and $|2\rangle$. We may expand the field operators, $\hat{\psi}_i^\dagger(\boldsymbol{r})$, for atoms in $|1\rangle$ using Wannier functions, $w(\boldsymbol{r} - \boldsymbol{r}_\text{s})$, as we did before in section 2. But, since we're assuming that $|2\rangle$ is unoccupied, we can no longer expand the field operators for $|2\rangle$ using the TF wavefunction. Instead, we will

expand $\hat{\psi}_i^\dagger(\boldsymbol{r})$ using the basis of the 3D HO (single-particle) eigenfunctions, $\psi_n(\boldsymbol{r})$. Since, in general,

$$\int \mathrm{d}\boldsymbol{r}\,\psi_n(\boldsymbol{r})w(\boldsymbol{r}-\boldsymbol{r}_s) \neq 0,$$

atoms in $|1\rangle$, located in a lattice site at $\boldsymbol{r}_s$, will couple to many of the HO levels, which we will need to take into account to accurately model the dynamics.

As before, we can substitute Eqs. (2) and (3) into (1), expand the field operators, and then integrate over all space to obtain the Bose–Hubbard Hamiltonian. This time, we will need to sum over all of the 3D HO eigenstates that an atom in $|2\rangle$ may occupy, and the two lattice sites that an atom in $|1\rangle$ may occupy (see Fig. 5). The result is

$$\hat{H} = \sum_n \left[ \left( \hbar\omega_n + U_{02}^n N \right)\hat{n}_n - \hbar\left( \Omega_{1n}\hat{a}_n^\dagger\hat{\sigma}_- + \Omega_{1n}'\hat{a}_n^\dagger\hat{\sigma}_-' + \text{h.c.} \right) \right]$$
$$+ \left( \hbar\Delta + U_{01}N \right)\hat{\sigma}_1 + \left( \hbar\Delta' + U_{01}'N \right)\hat{\sigma}_1', \tag{11}$$

where the summation over n is over all of the HO levels, $\hat{\sigma}_1$ and $\hat{\sigma}_1'$ are the number operators for atoms in each lattice site, $\hat{n}_n$ is the number operator for the $n^{\text{th}}$ HO energy level, $\Omega_{1n}$ and $\Omega_{1n}'$ are the Rabi frequencies for the transitions between each lattice site and the $n^{\text{th}}$ level, $\hbar\omega_n$ is the energy of the $n^{\text{th}}$ level, $U_{02}^n$ is the interaction energy between an atom in $|0\rangle$ and one in the $n^{\text{th}}$ level, $U_{01}$ and $U_{01}'$ are the interaction energies for atoms in each lattice site with those in $|0\rangle$, and $\Delta$ and $\Delta'$ are the detunings of the $\Omega_{12}$ field from resonance for each lattice site. As in Eq. (5), we assumed that each lattice site could contain at most one atom. Additionally, we neglected to account for interactions between lattice atoms and atoms in the HO due to the fact that there are at most two atoms participating in the dynamics, and thus the interaction terms we dropped will be many orders of magnitude smaller than all other energies in the problem. Note that $\Omega_{1n}$, $\Omega_{1n}'$, $U_{02}^n$, $U_{01}$ and $U_{01}'$ may all be calculated using Eq. (6) by replacing $\phi(\boldsymbol{r})$ with $\psi_n(\boldsymbol{r})$.

Following the same steps as in section 3.2, we calculate an effective Hamiltonian that acts on the manifold of Fock states with zero atoms in the HO. This is a good approximation, provided $\Delta - \omega_n \gg \Omega_{1n}, \Omega_{1n}'$. Using Eq. (8), we obtain the effective Hamiltonian

$$H_{\text{eff}} = \hbar \sum_n \Omega_{1n}\Omega_{1n}' \left( \frac{1}{\bar{\Delta} - \bar{\omega}_n} + \frac{1}{\bar{\Delta}' - \bar{\omega}_n} \right) (\hat{\sigma}_+\hat{\sigma}_- + \text{h.c.})$$
$$+ \hbar\left( \bar{\Delta} + \sum_n \frac{\Omega_{1n}^2}{\bar{\Delta} - \bar{\omega}_n} \right)\hat{\sigma}_1 + \hbar\left( \bar{\Delta}' + \sum_n \frac{\Omega_{1n}'^2}{\bar{\Delta}' - \bar{\omega}_n} \right)\hat{\sigma}_1', \tag{12}$$

where $\bar{\Delta} = \Delta + U_{01}N/\hbar$, $\bar{\Delta}' = \Delta' + U_{01}'N/\hbar$, and $\bar{\omega}_n = \omega_n + U_{02}^n N$. The first term in Eq. (12) leads to coupling between the two qubits, the second two modify the qubit energies slightly. We may eliminate (to first order) the small energy difference between the two qubits by tuning $\bar{\Delta}$ and $\bar{\Delta}'$, which leaves us with just the coupling term. Up to phase factors, Eq. (12) is then equivalent to the $\sqrt{\text{SWAP}}$ gate operation if we keep the field on just long enough to realize a $\pi/2$ pulse between $|\!\downarrow\uparrow\rangle$ and $|\!\uparrow\downarrow\rangle$. We may then apply a phase shift to the qubits to realize the $\sqrt{\text{SWAP}}$ gate.

After summing over the lowest 700 nondegenerate levels of the 3D HO (corresponding to $57,657,951$ individual states), we found that the coupling term in Eq. (12) was too small to yield reasonable gate times. This is consistent with previous work on this type of method for transferring atoms between traps using a spin-dependent potential, where it was found that the approach fails when $|\bar{\Delta}| \gg |\bar{\omega}_n - \bar{\omega}_{n-1}|$ for all n [67].

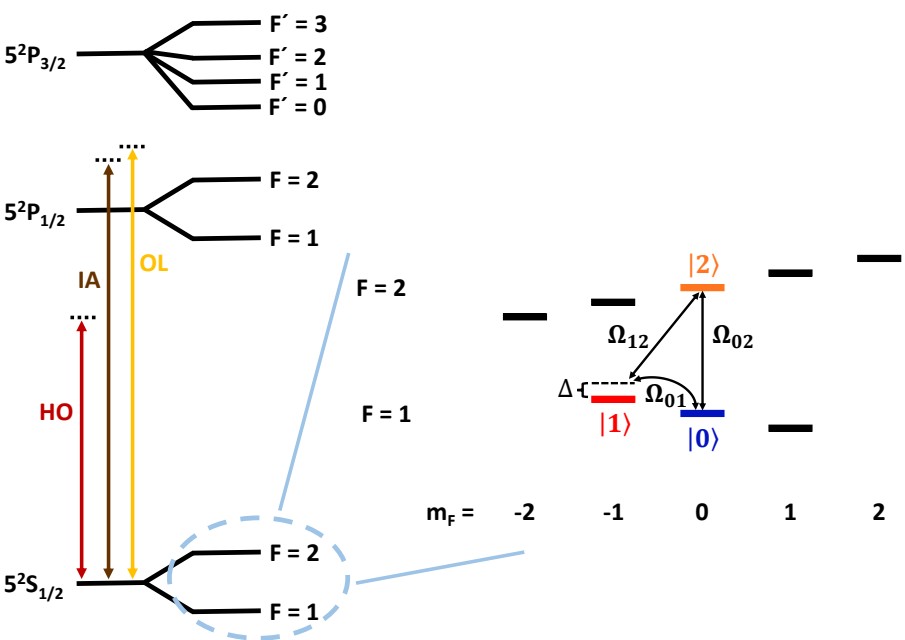

Figure 6: The $^{87}$Rb level spectrum. The far-detuned optical beams are shown on the left. The HO beams create the harmonic trap, the OL beams the lattice, and the IA beams are used to individually address the qubits. The three levels in the $5^2S_{1/2}$ ground state manifold and coupling fields that were chosen to realize the model system in section 2 are shown on the right.

To rectify this, we apply an $\Omega_{02}$ field far from resonance to couple the empty HO states in $|2\rangle$ to the BEC in $|0\rangle$. In this limit, the HO levels, $\bar{\omega}_n$, will shift according to Eq. (8) (the AC Stark effect) due to the $\Omega_{02}$ field. The the $\hat{\Omega}$ field terms in Eq. (8) may be calculated using Eq. (6) by replacing $w(r - r_0)$ with $\psi_n(r)$. We found that using the spatial profile $\hat{\Omega}(r) \propto e^{-(x^2+y^2)/w^2}$ for the field works well. This profile for $\Omega_{02}$ was chosen because it is the approximate spatial profile of a Gaussian laser propagating along the z-axis with a beam waist of $w$, and is thus reasonable to realize experimentally. The effect of this field is to shift the HO levels such that $|\bar{\Delta}| < |\bar{\omega}_n - \bar{\omega}_{n-1}|$ for certain n, thereby allowing the gate time to be made reasonable (see section 4.2).

## 4 Experimental realization

### 4.1 The platform

Thus far, we have discussed a simplified model of bosons confined to a spin-dependent lattice and harmonic potential, which we showed could be used as a quantum register capable of universal quantum computation. In this section, we consider how the model system could be well approximated using an experimentally realistic setup, and confirm that the realistic system will behave analogously to the model one using computations.

Alkali atoms are workhorses of atomic physics experiments, and although in our computations we consider the particular case of $^{87}$Rb atoms, we note that the model in section 2 could also be applied to other bosonic alkali species [68, 69]. In order to realize the Hamiltonian given by Eq. (5), we must first trap the atoms in a potential that resembles Eq. (3), which is usually done using optical fields [40–42, 68, 70].

A neutral atom in an optical field that is far-detuned from any resonances can be treated

as an electric dipole [71]. If the oscillating electric field is a perturbation, and only the $^2P_{1/2}$ and $^2P_{3/2}$ fine structure levels of alkali atoms are taken into account (see Fig. 6), then to second-order in perturbation theory the conservative potential due to the AC Stark shift felt by an alkali atom in its ground state will be [71]

$$V_{\text{HO}}(\boldsymbol{r}) = \frac{\pi c^2 \Gamma}{2\omega_0^3} \left( \frac{2 + P g_{\text{F}} m_{\text{F}}}{\Delta_2} + \frac{1 - P g_{\text{F}} m_{\text{F}}}{\Delta_1} \right) I(\boldsymbol{r}), \tag{13}$$

where $c$ is the speed of light, $\Gamma$ is the spontaneous emission rate of the excited state, $\omega_0$ is the transition frequency, $P = 0, \pm 1$ denotes whether the light is linearly or $\sigma^\pm$ circularly polarized, $g_{\text{F}}$ is the Landé g-factor, $m_{\text{F}}$ denotes which Zeeman sublevel the atom is in, $\Delta_1$ is the detuning of the optical field from the $^2S_{1/2} \rightarrow {}^2P_{1/2}$ transition, $\Delta_2$ is the detuning from the $^2S_{1/2} \rightarrow {}^2P_{3/2}$ transition, and $I(\boldsymbol{r})$ is the intensity of the beam of light. Note that the $\Gamma/\omega_0^3$ factor is only approximately the same for the two transitions [72].

The isotropic harmonic trapping potential considered in Eq. (3) may be achieved using two crossed optical beams with Gaussian intensity profiles [71]. If $\Delta_1, \Delta_2 < 0$ (red-detuning, see the HO beams in Fig. 6), then the atoms will be attracted to areas of greater intensity. Thus, near the intersection point of the two beams (where the total intensity is greatest), the atoms will see an approximately harmonic trapping potential. And if both beams are linearly polarized ($P = 0$), then all atoms in the $^2S_{1/2}$ manifold will see approximately the same harmonic potential. This assumes that any interference effects will be negligible (i.e. that the dipole potentials due to each beam, calculated using Eq. (13), may be added together).

In our computations, we created the isotropic harmonic trap using two crossed $\lambda = 808$ nm beams, each with a total power of 100 mW. One beam had waists of 120 μm and 90 μm, while the other had 132 μm and 87.3 μm waists. These result in a harmonic trapping frequency of $\omega_{\text{HO}} = 2\pi \times 100$ Hz.

To realize the 3D spin-dependent lattice potential part of Eq. (3), we follow [40] and consider three pairs of nearly counterpropagating (Gaussian) beams. Each beam is linearly polarized, and one beam in each pair has had its polarization vector rotated by an angle $\theta$ with respect to its partner beam. In this case, the dipole potential due to each pair of beams is given by [40]

$$V_{\text{OL}}(\boldsymbol{r}) = \frac{\pi c^2 I_0 \Gamma}{\omega_0^3} \left\{ \left( \frac{2}{\Delta_2} + \frac{1}{\Delta_1} \right) (1 + \cos(\theta) \cos(2\boldsymbol{k} \cdot \boldsymbol{r})) \right. $$
$$\left. + g_F m_F \left( \frac{1}{\Delta_2} - \frac{1}{\Delta_1} \right) (\hat{\boldsymbol{k}} \cdot \hat{\boldsymbol{B}}) \sin(\theta) \sin(2\boldsymbol{k} \cdot \boldsymbol{r}) \right\}, \tag{14}$$

where $I_0$ is the intensity at the center of each beam at its waist, $\boldsymbol{k}$ is the wavevector of each beam, $\boldsymbol{B}$ is the magnetic field, and $\hat{\boldsymbol{k}}$ and $\hat{\boldsymbol{B}}$ are the wavevector and magnetic field's unit vectors. Note that $|\boldsymbol{k}| = 2\pi/\lambda$, where $\lambda$ is the wavelength of each beam.

Eq. (14) describes the sinusoidal dipole potential due to two counterpropagating beams if both beams have the same frequency, and thus interfere with each other. To create the spin-dependent lattice used in section 2, we require two atomic states that will see no lattice potential ($|0\rangle$ and $|2\rangle$), and one state that does see a lattice ($|1\rangle$). To achieve this using Eq. (14), we select $|0\rangle = |F = 1, m_{\text{F}} = 0\rangle$, $|1\rangle = |F = 1, m_{\text{F}} = -1\rangle$, and $|2\rangle = |F = 2, m_{\text{F}} = 0\rangle$ (see Fig. 6). We also set $\theta = 90°$, and align the magnetic field, $\boldsymbol{B}$, so that its components are equal along all three lattice directions. In this case, Eq. (14) becomes (up to a constant)

$$V_{\text{OL}}(\boldsymbol{r}) = \frac{\pi c^2 I_0 \Gamma}{\sqrt{3}\omega_0^3} g_{\text{F}} m_{\text{F}} \left( \frac{1}{\Delta_2} - \frac{1}{\Delta_1} \right) \sin(2\boldsymbol{k} \cdot \boldsymbol{r}). \tag{15}$$

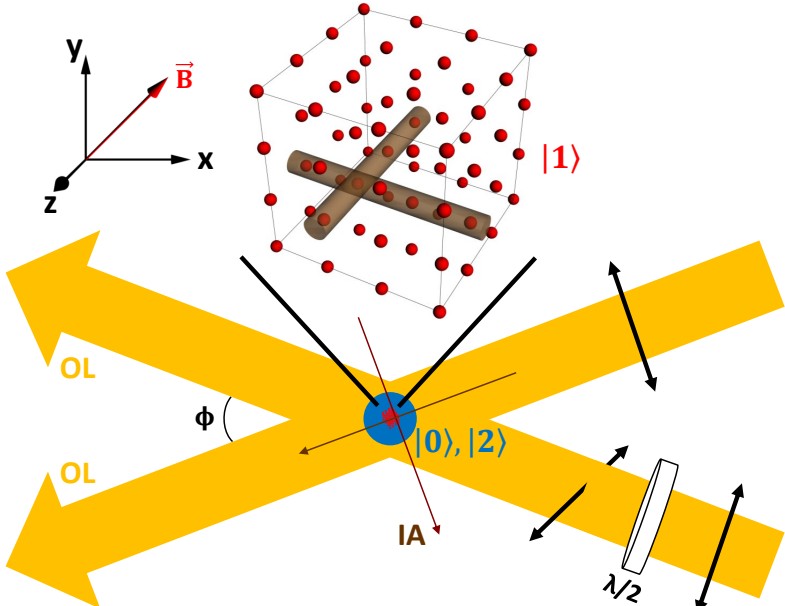

Figure 7: Experimental setup. The pair of OL beams that create the lattice potential along the y-axis are shown intersecting at an angle $\phi$. One beam passes through a half-wave plate and has its polarization vector rotated by $\theta = 90°$. The atoms in states $|0\rangle$ and $|2\rangle$, which see an isotropic harmonic potential but no lattice, are shown having condensed into a dilute BEC. Atoms in $|1\rangle$, which do see the 3D lattice potential, are tightly trapped in a cubic array. The two tightly focused, crossed beams that are used to individually address atoms in the lattice are labelled IA.

With three pairs of beams propagating along orthogonal directions a 3D lattice potential is obtained. $|0\rangle$ and $|2\rangle$ will see the potential $V_{\text{HO}}$ only, while $|1\rangle$, which has $m_{\text{F}} \neq 0$, will see the total potential $V_{\text{HO}} + V_{\text{OL}}$, which is consistent with Eq. (3). Note that we are restricted in the choice of wavelength, $\lambda$, because it must be close to the ${}^2\text{P}_{1/2}$ and ${}^2\text{P}_{3/2}$ transitions. Otherwise, $1/\Delta_2 - 1/\Delta_1$ in Eq. (15) will be too small and the intensities of the beams would need to be unreasonably large.

The arrangement of beams discussed above is shown in Fig. 7. Note that if each pair of beams intersect at an angle $\phi$, then in Eq. (15) $\mathbf{k}$ will be changed to $\mathbf{k} \sin(\phi/2)$ [40, 73, 74]. By tuning $\phi$ the effective lattice spacing may be increased, but to maintain consistency with the assumptions in section 2 the lattice spacing must be much less than the length scale of the HO.

In the computations, the 3D optical lattice was formed by 3 pairs of nearly counterpropagating $\lambda = 790$ nm beams. This 'magic' wavelength was chosen so that atoms in $|0\rangle$ and $|2\rangle$ won't see the lattice potential even if the beams are imperfectly polarized (because $2/\Delta_2 + 1/\Delta_1 = 0$ in Eq. (14)). Each beam had a total power of 67 mW, and a beam waist of 150 μm in each direction, which leads to a lattice depth of 800 kHz. We only consider the $n = 10 \times 10 \times 10 = 1,000$ lattice sites closest to the trap center to use as qubits.

Initializing a gas of bosonic atoms into a BEC has two main steps: laser cooling and forced evaporative cooling [50, 75–78]. A common way to then prepare all of the atoms in a single Zeeman sublevel is to turn on a purely magnetic gradient with a local minimum, in which case only atoms with $m_{\text{F}} < 0$ will see a trapping potential (due to the Zeeman effect), and atoms with $m_{\text{F}} \geq 0$ will be expelled from the trapping region. It is also possible to instead use optical pumping lasers to force all of the atoms into a single spin state [79]. Once every atom is loaded into the $|0\rangle$ state, the qubit register will have been initialized with every qubit in $|\downarrow\rangle$.

Note that in our system the lattice does not need to be initialized with one atom in each site before computations can begin, which is not the case in other lattice-based qubit platforms. Typically, initialization requires loading exactly one atom into each site. This is not trivial because loading schemes usually rely on atoms in the same site colliding, leading both to be ejected from the trap. This results in each lattice site randomly having zero or one atom(s). Thus, qubit registers usually need to be initialized by then rearranging the atoms in the lattice so that a subsection of the lattice has perfect unit filling [80–82].

In our computations, we initialized our system with $10^6$ $^{87}$Rb atoms at $T = 300$ nK (which is representative of the number of atoms that can be loaded into a harmonic trap at this temperature [40, 54, 60]). The BEC fraction will be about 70%, which means that there will be $N = 7 \times 10^5$ BEC atoms in the trap (and $3 \times 10^5$ noncondensed, thermal atoms).

The scattering lengths, $a_{ij}$, in Eq. (2) for atoms in states $|0\rangle$, $|1\rangle$ and $|2\rangle$ are $a_{00} = 5.34$ nm, $a_{11} = 5.31$ nm, $a_{22} = 5.00$ nm, $a_{01} = 5.31$ nm, $a_{02} = 5.23$ nm, and $a_{12} = 5.16$ nm [46–48]. For the given choice of the lattice and harmonic trap beam parameters, the corresponding interaction strengths in Eq. (5) will be $U_{00}/\hbar = 0.0197$ Hz, $U_{11}/\hbar = 13.0$ kHz, $U_{22}/\hbar = 0.0184$ Hz, $U_{01}/\hbar = 0.0342$ Hz, $U_{12}/\hbar = 0.0332$ Hz, and $U_{02}/\hbar = 0.0193$ Hz.

## 4.2 Qubit gates

The second step to implement the model Hamiltonian in Eq. (5) is to realize the coupling terms between the atomic states: $\Omega_{01}$, $\Omega_{12}$, and $\Omega_{02}$. As shown in Fig. 6, $\Omega_{01}$ connects two Zeeman sublevels in the same hyperfine manifold. If we apply a uniform magnetic field, $|\boldsymbol{B}| = 5.40$ G, then the energy difference between $|0\rangle$ and $|1\rangle$ will be 3.78 MHz [72]. State $|2\rangle$ is in a different hyperfine manifold than $|0\rangle$ and $|1\rangle$, which makes the energy difference between these states much larger (6.835 GHz) [72].

The transitions between these three atomic states could either be driven using radiofrequency (RF) and microwave (MW) fields, or with two coherent optical beams that are detuned from each other by exactly the energy difference between the two states being connected (thereby driving a Raman process) [58–60]. The most common methods of addressing an individual qubit in a lattice while using MW and RF fields is to either generate the fields using an atom chip [68, 83, 84], or apply tightly focused optical beams to preferentially shift the energy levels of the addressed qubit relative to its neighbors (see Fig. 7) [85–87]. Driving these transitions with optical beams would have the advantage of not requiring an atom chip or any additional tightly focused optical beams, but driving a Raman transition would increase the rate of atom losses in the system (see section 5.1).

The intersection angle of the lattice beams, $\phi$, was chosen so that the lattice spacing will be 532 nm, which is sufficient to allow the lattice sites to be individually addressed by tightly focused optical beams [85–87] (although alternative approaches for addressing individual sites exist [88]). The individual addressing beams were chosen to have $\lambda = 790$ nm with beam waists of 0.6 μm and circular polarization. This 'magic' wavelength guarantees that atoms in $|0\rangle$ and $|2\rangle$ won't see the addressing beams, but atoms in $|1\rangle$ will. It is assumed that the addressing beams can be rapidly reconfigured to focus on any qubit in the 3D lattice, for instance, by using a lens on a translation stage and MEMS mirrors as in [86, 87, 89].

Given the harmonic trap and lattice potential described in section 4.1, we obtain $\phi(\boldsymbol{r})$ [50, 54] and $w(\boldsymbol{r})$ [49, 51, 53]. Then, using Eq. (6), we find $\Omega_{ij} = \tilde{\Omega}_{ij}/364$, where $\tilde{\Omega}_{ij}(\boldsymbol{r})$ in Eq. (6) was assumed to be a square pulse with amplitude $\tilde{\Omega}_{ij}$. The effective Rabi frequency for single-qubit gates is $\sqrt{N_0}\Omega_{ij}$ (from Eq. (7)), and since $\sqrt{N_0} \sim 837$, we find that the effective Rabi frequency will be about $837/364 = 2.30$ times larger than the free-space Rabi frequency, $\tilde{\Omega}_{ij}$. Smaller field strengths will therefore be needed to drive transitions in our system compared to an analogous single-atom system, which is one of the advantages of ensemble qubits. We note, however, that this enhancement is significantly smaller than in most ensemble qubits

(due to the small overlap between $\phi(\boldsymbol{r})$ and $w(\boldsymbol{r})$) [10–14, 32]. In our computations, we chose $\sqrt{N_0}\Omega_{01} = 2\pi \times 100$ Hz, which yields a $\pi$-pulse time of 5.03 ms. This was computed for qubits farthest from the center of the trap (i.e. qubits for which $\Omega_{01}$ is smallest, and thus the gate time will be longest).

In sections 3.2 and 3.3 we considered using the BEC, or an empty mode of the HO, as a native quantum bus to entangle distant pairs of qubits. This approach is advantageous compared to many other two-qubit gate protocols in that it does not require any of the atoms to be shuttled around [3, 15], brought outside of the qubit basis [22], or an optical cavity [18].

To realize the CNOT gate discussed in section 3.2, we chose $\Omega_{01} = \Omega_{12} = 2\pi \times 192$ Hz, and $\Delta = 12.9$ kHz. For the first step of the protocol, where we couple the BEC to the control qubit (Fig. 2), the effective Rabi frequency will be $\Omega_{01}\Omega_{12}/\Delta = 2\pi \times 18.0$ Hz, with a pulse time of 11.1 ms. The second step is to perform a $\pi$-pulse on the target qubit (third column of Fig. 4), which will take 5.03 ms. And the last step is to reset the BEC bus, which will take another 11.1 ms, yielding a total CNOT gate time of $\sim 27.2$ ms.

We note that the CNOT time is primarily limited by the small overlap between $\phi(\boldsymbol{r})$ and $w(\boldsymbol{r})$ in Eq. (6), and that reducing this gate time would require larger free-space Rabi frequencies, $\tilde{\Omega}_{01}, \tilde{\Omega}_{12} > 1$ MHz. It is also worth noting that unlike Rydberg-based schemes for realizing a CNOT gate [4], using the BEC as a quantum bus means that we cannot perform the gate operation on multiple pairs of qubits in parallel.

In the first step of the CNOT protocol, a single qubit is entangled with the BEC. This may be done using either an atom chip [68, 83, 84], or a pair of tightly focused optical beams that drive a Raman transition between $|0\rangle$ and $|2\rangle$ [58–60]. In the second step a single-qubit gate must be performed on a different qubit in the lattice, which may be done using an atom chip or rapidly reconfigurable optical beams [86, 87, 89].

To realize the $\sqrt{\text{SWAP}}$ gate proposed in section 3.3, we first note that any $^{87}$Rb atomic level with $m_F = 0$ could be substituted for state $|2\rangle$. We chose the $|F = 1, m_F = 0\rangle$ level of the $5^2P_{1/2}$ excited band in our computations because this allows us to use optical fields for $\Omega_{1n}$ and $\Omega'_{1n}$ in Eq. (12). To realize the correct spatial mode for $\Omega_{02}$ discussed in section 3.3, we applied a linearly polarized $\lambda = 532$ nm beam with a total power of 0.2 mW, and beam waists of 14.1 μm in each direction. Since optical fields are being considered, to address different pairs of qubits in the lattice we require two pairs of rapidly reconfigurable and tightly focused Raman beams.

Qubits whose lattice sites are mirror reflections of each other across the center of the harmonic trap will require exceptionally small field strengths to entangle compared to other qubit pairs with our method. For sites that are in opposite corners of the 3D lattice (and therefore have the weakest coupling to the modes of the HO), we found that $\tilde{\Omega}_{12} = 2\pi \times 11.2$ kHz and $\Delta = -2.14$ kHz led to an effective Rabi frequency of 19.0 kHz, and thus a gate time of 82.7 μs. $\tilde{\Omega}_{12}$ was chosen so that $\Omega_{1n}, \Omega'_{1n} \leq (\Delta - \omega_n)/10$ for all n, as required for Eq. (12) to be valid.

For pairs of sites that are not mirror reflections across the trap center, much larger field strengths are necessary. To connect the site in position $(-5, -5, -5)$ (i.e. the corner of the $10 \times 10 \times 10$ lattice), to the site at $(5, 2, 1)$ we required $\tilde{\Omega}_{12} = 2\pi \times 5.68$ MHz and $\Delta = -8.00$ MHz to achieve an 82.7 μs gate time.

## 4.3 Nondestructive state readout

We propose a method to read out the state of any individual qubit in our array by coupling the qubit to the BEC, and then nondestructively measuring the state of the BEC. This can be done using first step of the protocol given in section 3.2 (see Fig. 2). $\Omega_{01}$ and $\Omega_{12}$ are used to drive population between levels $|0\rangle$ and $|2\rangle$, which will be successful if and only if the qubit to be measured is in $|\downarrow\rangle$. The period of the BEC population oscillations would be 11.1 ms.

There are several methods to nondestructively detect population oscillations in a BEC. It was shown in [34] that Rabi oscillations between $|0\rangle$ and $|2\rangle$ in ${}^{87}$Rb atoms may be observed by measuring the MW impedance of the atoms. In [37], BEC atoms coherently exchanged photons with two optical beams, and the resulting intensity fluctuations in the beams were observed on a CCD camera. To address an individual qubit in the lattice, either an atom chip or a pair of tightly focused optical beams may be used (as discussed in section 4.2).

Compared to more conventional measurement approaches [25–27], in the one proposed here there is little chance of the qubit atoms being heated or lost during the imaging process, and no high-resolution optics are required. The qubit is also never removed from the computational basis. However, only one qubit can be measured at a time, and there will be decoherence introduced to the BEC during readout. We also note that the state of the qubits in our system could instead be read out with the usual fluorescence-based approaches; the method described here is just a second option not available to other lattice-based qubits.

# 5    Sources of Decoherence

## 5.1    Sensitivity to atom losses

One of the most important sources of error in cold atom qubit platforms are atom losses. In cold atom qubits where the quantum information is stored within a single atom [2–8], each atom loss completely destroys a qubit and is therefore a significant source of decoherence for these platforms.

There are, however, schemes to detect and correct such errors [90–92]. Atomic losses are less problematic for qubits made up of ensembles of atoms, because the loss of an atom does not mean that all of the quantum information contained in that qubit has been destroyed. Although, for ensemble qubits based on Rydberg excitations [4,10,13], the size of the ensemble is typically at most 100 atoms, which means atom losses in these systems are not entirely negligible either.

In our proposed qubit register, the qubit atoms will be drawn from an ensemble of $N$ atoms. The qubits states were defined to be an empty lattice site ($|\downarrow\rangle$) and a filled site ($|\uparrow\rangle$). This choice means that atom losses do not destroy our qubits, but rather represent an energy relaxation process that results in bit flips $|\uparrow\rangle \rightarrow |\downarrow\rangle$.

To quantify the effect of atom losses on the quantum information stored in our qubit register, we follow the approach in [12]. When $\eta < N$ atoms are lost from the system, the probability that a lattice atom was lost (which is the only type of atom loss that results in the destruction of information) can be found by calculating the ratio between the number of states in which at least one lattice atom was lost, and the total number of states.

If $m$ qubits in the register are in the state $|\uparrow\rangle$, then we can write out the state of all $N$ atoms by noting that each of the $N$ atoms is equally likely to have been excited to each of the $m$ lattice sites. The (unnormalized) superposition state is thus

$$|12...m00...0\rangle + |12...0m0...0\rangle + ...|00...012...m\rangle, \tag{16}$$

where $1, 2, ..., m$ denote which lattice site that atom is in, and $0$ denotes that the atom is in the harmonic trap. We observe that there are a total of $P(N, m)$ states in the superposition given by Eq. (16) (where $P(N, m)$ denotes $m$-permutations of $N$).

We consider what happens, without loss of generality, when the first $\eta < N$ atoms are lost. In $P(N - \eta, m)$ of the states in Eq. (16), the first $\eta$ atoms do not carry any quantum information. Thus, the probability of at least one lattice atom being lost, $P_\eta$, is equal to the probability that the system is not in one of those states,

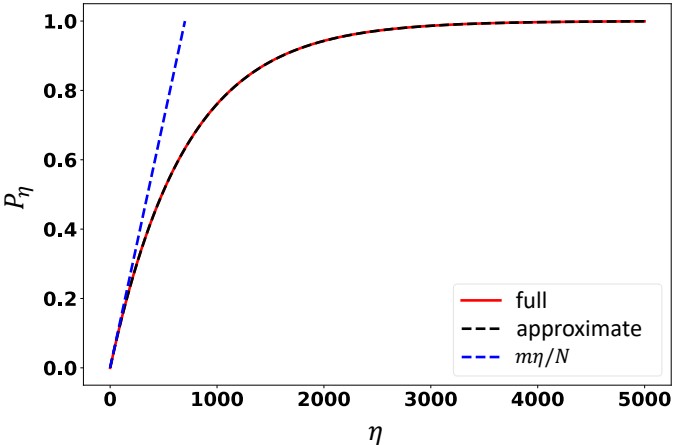

Figure 8: Probability of a bit flip occurring, $P_\eta$, as a function of the number of atom losses, $\eta$. The curve labeled 'full' corresponds to Eq. (17), and 'approximate' to Eq. (18). $N = 7 \times 10^5$ atoms and $m = 1,000$ qubits in the $|\uparrow\rangle$ state were used to make the plot.

$$P_\eta = 1 - \frac{P(N-\eta, m)}{P(N, m)} = 1 - \prod_{j=0}^{m-1}\left(1 - \frac{\eta}{N-j}\right). \tag{17}$$

For $N \gg m$,

$$P_\eta \approx 1 - \left(1 - \frac{\eta}{N}\right)^m. \tag{18}$$

For $\eta \ll N$, we have $P_\eta \approx m\eta/N$, where $m\eta/N$ is the expected number of compromised qubits after $\eta$ losses.

$P_\eta$ is plotted in Fig. 8 using $N = 7 \times 10^5$ atoms, and $m = 10^3$ qubits in $|\uparrow\rangle$ (i.e. every qubit is excited). We observe that, as compared to platforms where each qubit is made up of a single atom, in the scheme proposed here the quantum register will be well protected against atom losses that are much smaller than the size of the ensemble.

In the proposed system, we expect losses will primarily arise from atoms scattering photons from the optical lattice beams (expected lifetime of 0.522 sec) [71], heating due to momentum diffusion in the spin-dependent lattice (4.99 sec) [40], collisions with the background gas (5.0 sec) [93, 94], and three-body losses in the BEC (18.3 sec) [45].

## 5.2 Coupling to states outside the computational basis

In the simple model introduced in section 2, atoms can occupy only three states: atoms in $|0\rangle$ and $|2\rangle$ occupy the many-body ground state of the harmonic trap, and atoms in $|1\rangle$ lie in the lowest Bloch band of one of the lattice sites. In the realistic system described in section 4, however, there are more states in the Hilbert space than just three, which can lead to qubits being forced out of the computational basis (compare Figs. 1 and 9). This can happen if multiple atoms occupy a single lattice site, or if atoms in the lattice transition to higher-lying HO states, other Bloch bands, or other atomic levels. In this section, we explore the primary ways by which these can happen.

For a deep optical lattice, there will be strong s-wave interactions between atoms in the same site, $U_{11}/\hbar = 13.0$ kHz. While performing the qubit operations described by Eq. (7), we therefore require $\sqrt{N_0}\Omega_{01} \ll U_{11}/\hbar$ to avoid double occupation of the lattice site. This is the most stringent constraint on the single-qubit and CNOT gate speeds of our system. If two

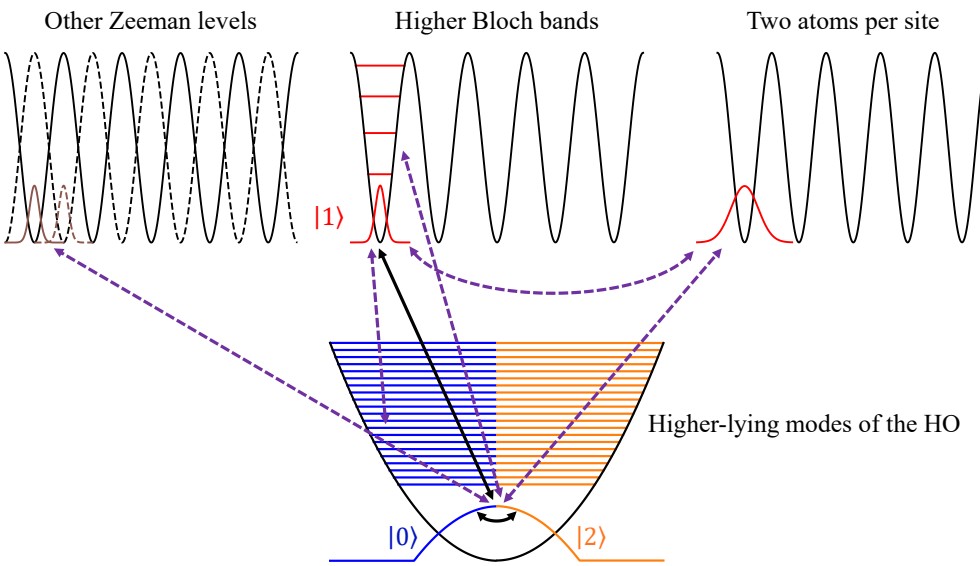

Figure 9: Coupling between the computational basis states and other states. The three basis states in the simplified model in Fig. 1 are shown, along with their coupling to states that lie outside this basis. Desired couplings that are used to perform qubit operations are shown with solid arrows, all undesired couplings are displayed with dashed arrows. Atoms in the BEC couple to other Zeeman sublevels, higher Bloch bands in the optical lattice, and states with multiple atoms per site. Qubit atoms in the lattice may couple to the excited levels of the harmonic trap, or multiply filled sites due to inter-site tunneling.

atoms do end up sharing a single site, then they will most likely collide and both be ejected from the trap within one millisecond [93, 94]. If the second atom came from the BEC or the background gas of thermal (noncondensed) atoms, then this is an energy relaxation process that maps $|\uparrow\rangle \rightarrow |\downarrow\rangle$.

It is also possible for atoms to tunnel through the potential barrier between lattice sites. If an atom tunnels into another, occupied, lattice site, then the two atoms will collide and both will be lost from the trap. Tunneling therefore maps $|\uparrow\uparrow\rangle \rightarrow |\downarrow\downarrow\rangle$ if both sites were initially occupied, or $|\uparrow\downarrow\rangle \rightarrow |\downarrow\uparrow\rangle$ if only one site was occupied. The lattice beam parameters given in section 4.1 lead to a tunneling energy of 0.09 Hz [50–52]. Tunneling between lattice sites could be suppressed by using a deeper lattice, or by applying a potential gradient [95] (i.e. by aligning the lattice with gravity, applying a magnetic field, or accelerating the lattice).

We defined an empty lattice site to be $|\downarrow\rangle$, and because of this, when multiple atoms occupy the same lattice site, the system will eventually return to the computational basis of its own accord. This may be compared to the situation in many other lattice-based qubits, where atoms losses must be manually detected and corrected [90–92]. This inherent protection against atoms losses (see section 5.1) comes at the cost of those losses being more frequent in our system because the BEC and thermal atoms must be co-trapped with the lattice atoms, leading to increased opportunities for them to collide with each other and be lost from the trap.

In addition to collisions between atoms, during qubit operations we must also contend with atoms being transferred to both the higher-lying levels of the HO, and the higher Bloch bands of the lattice. This population transfer occurs because the spatial wavefunctions of atoms in the lattice ($w(\mathbf{r})$) overlap with the higher-lying modes of the harmonic trap, and the many-body wavefunction of the BEC atoms ($\phi(\mathbf{r})$) also overlaps with the spatial modes of the higher Bloch bands in the lattice (see Fig. 9).

The thermal atoms will be distributed over the excited levels of the 3D HO according to the Bose–Einstein distribution [36, 50, 54]. The coupling strengths between the qubit states and these excited levels will be proportional to $\sqrt{N_e}$, where $N_e$ is the number of thermal atoms in the excited level e. According to the Bose–Einstein distribution, the first excited level will have the largest population, $N_e = 62$ atoms. Since $N_0 = 5 \times 10^5$, the coupling strength between the qubit levels will be approximately $\sqrt{N_0/N_e} = 106$ times larger than the maximum coupling strength to the first excited level of the HO.

To minimize the coupling to the higher Bloch bands of the lattice, $\sqrt{N_0}\Omega_{01}$ and $\sqrt{N_2}\Omega_{12}$ must be much less than the frequency difference between Bloch bands (which is 190 kHz for our lattice [50, 51]). This is the primary factor which limits the speed of the $\sqrt{\text{SWAP}}$ gate. Unlike the single-qubit and CNOT gates, which are constrained to have Rabi frequencies $\ll U_{11}/\hbar = 13.0$ kHz to avoid multiple atoms occupying the same lattice site, the $\sqrt{\text{SWAP}}$ gate does not require population to be transferred into and out of the lattice, and can therefore have a Rabi frequency of up to $\sim 19.0$ kHz. Coupling to the excited levels of the HO and coupling to the higher bands of the lattice are both drawbacks that are unique to our proposed system. These are not issues faced by most other lattice-based qubits, because on those platforms the spatial mode of the qubits is usually not changed significantly during qubit operations.

The $\Omega_{01}$, $\Omega_{12}$, and $\Omega_{02}$ fields are used to drive transitions between $|0\rangle$, $|1\rangle$ and $|2\rangle$. As can be seen in Fig. 6, the ground state manifold of $^{87}$Rb contains five other Zeeman sublevels, all of which could in principle be populated while we perform qubit operations. Spin-changing collisions between pairs of atoms can also cause undesired population transfer to other Zeeman levels [40, 46–48, 64]. To prevent either of these from happening, a moderate magnetic field, $\boldsymbol{B}$, must be applied so that the energy spacing of the Zeeman sublevels will be anharmonic, allowing us to only drive the transitions shown in Fig. 6. For $|\boldsymbol{B}| = 5.4$ G there will be an anharmonicity of over 13 kHz [64].

We are also aided by the fact that three out of the five undesired Zeeman levels will see a lattice potential that is 180° out of phase with the lattice potential seen by the atoms in $|1\rangle$ (because, for atoms in those states, $g_F m_F$ in Eq. (14) will have the wrong sign). Direct coupling between $|0\rangle$, $|1\rangle$ and $|2\rangle$ with these three levels will therefore be severely reduced before the nonlinear Zeeman effect is even taken into account. This means that $|1\rangle$ is not directly coupled to any of the undesired Zeeman levels, and $|0\rangle$ and $|2\rangle$ are only directly coupled to $|F = 2, m_F = 1\rangle$. However, in most cold atom qubit platforms, only one or two of the Zeeman sublevels in the ground state manifold are used to encode the quantum information. This makes population transfer out of the desired basis a much more significant problem in our case, although we note that it has been demonstrated that composite pulses may be used to reduce population leakage [96].

## 5.3 Decoherence due to the BEC and thermal atoms

Using the BEC to mediate qubit operations will introduce noise into the system. All of the error channels discussed in this section are due to co-trapping the BEC and thermal atoms with the qubit atoms in the lattice. Consequently, none of these sources of decoherence are present in other lattice-based qubits.

There are several impacts of the fact that we can only have a finite number of atoms, $N = N_0 + N_1 + N_2$, in the system. First, the Rabi frequency for qubit operations (the off-diagonal term in Eq. (7)) goes as $\sqrt{N_0}$. For trapped atoms, there will be fluctuations in $N$ (and thus $N_0$) between experimental realizations that go as $\sqrt{N}$ because it's a Poisson process [97]. In order for the Rabi frequency in Eq. (7) to be robust against these shot-to-shot fluctuations in $N$, we require $N >\sim 600$ atoms. The qubit transition frequency (the difference between the diagonal terms in Eq. (7)) will also vary due to these $\sqrt{N}$ shot-to-shot fluctuations. For $N = 7 \times 10^5$ atoms, there will be an uncertainty of $\leq 0.376$ Hz in each qubit's Rabi frequency,

and $\leq 28.7$ Hz in each qubit's transition frequency (these will be largest for the qubits closest to the center of the harmonic trap).

The CNOT gate, which is described by a Hamiltonian whose elements are given by Eq. (8), will also be sensitive to fluctuations in $N$. The first step of the CNOT gate is to initialize the BEC with $N_0 = N/5$ and $N_2 = 4N/5$, which cannot be done perfectly given our limited knowledge of $N$. If, as suggested in section 3.2, we apply the field $\Omega_{02} = 1$ kHz to drive population between $|0\rangle$ and $|2\rangle$, then $\sqrt{N}$ fluctuations between realizations will introduce an uncertainty of $2.11 \times 10^{-7}$ in the fraction of atoms in $N_0$ and $N_2$. The next step of the CNOT gate is to entangle the BEC with the first qubit. For an uncertainty in $N$ of $\sqrt{N}$, the error in the fractions of BEC atoms in $|0\rangle$ and $|2\rangle$ during this gate step will be 1.31%. The last step of the CNOT gate wil have the same uncertainty due to fluctuations in $N$ between realizations as the single-qubit gate discussed already.

Similar to the CNOT gate, there will be errors in implementing the $\sqrt{\text{SWAP}}$ gate discussed in section 4.2 due to shot-to-shot fluctuations in $N$. In Eq. (12), the energies of the initial, intermediate, and final states (see Fig. 5) depend on $N$. Fluctuations in $N$ of $\sqrt{N}$ will cause an uncertainty in the initial and final states' energy difference of about 0.03 Hz, and in the effective Rabi frequency of about 0.01% (these numbers will differ slightly for different pairs of qubits).

Decoherence due to finite $N$ also arises because $N_0$ and $N_2$ will vary slightly depending on how many qubits in the register are in the state $|\uparrow\rangle$. If there are $n$ qubits in our register, then we require $N \gg n$, otherwise the parameters of each qubit will depend on the state of every other qubit. Since, in our proposal, $n \approx \sqrt{N}$, the decoherence from this source will be similar to the decoherence due to shot-to-shot fluctuations in $N$ discussed above. We also note that there are many general techniques for minimizing the impact of fluctuating or unknown qubit parameters [98–103].

The final drawback of finite $N$ is that atom losses in the BEC will lead $N$ to decrease over time, which in turn will cause each qubit's transition frequency, and sensitivity to the driving fields, to vary in time (see section 5.1). If the loss rate of BEC atoms over time can be estimated, then the decoherence introduced to the system from this error channel may be minimized by correcting for these losses when setting the gate times and field frequencies. Additionally, techniques to continuously reload a BEC to counter the effects of atom losses have already been demonstrated experimentally [104].

There are several approaches which could be used to reduce the decoherence introduced by uncertainty in $N$. The interactions between the BEC atoms and all other atoms in the system could be reduced by decreasing the BEC density. However, this would also decrease the gate speed. In [105], hundreds of ensembles of atoms with sub-Poissonian atom number fluctuations were created. And in [106], the authors proposed a method to estimate the size of an atomic ensemble using multiple pulses and measurements of Rydberg excitations.

During the CNOT gate, BEC population is transferred between $|0\rangle$ and $|2\rangle$. This will leave the spatial state of the atoms unchanged, which is problematic because the change in the population of atoms in different atomic states will cause their spatial states to no longer be eigenstates of the Hamiltonian. As a result, during the CNOT gate operation, the wavefunctions of atoms in $|0\rangle$ and $|2\rangle$ will evolve in time. Uncertainty in the gate time will therefore result in uncertainty in the qubit parameters in Eq. (7). We simulated these dynamics using a fourth-order Runge–Kutta method to solve the Gross–Pitaevskii equation for a spinor BEC [50, 54]. It was found that, for an uncertainty in pulse length of 1.81 μs, the error in the effective Rabi frequency should always be less than 0.001.

The change in populations of $|0\rangle$ and $|2\rangle$ will also change the condensation temperature, $T_c$ of atoms in these two states, since $T_c \propto N^{1/3}$ [50, 54]. Any decrease in $T_c$ due to population transfer will increase the thermal component of the BEC [40]. During the CNOT gate,

the population of one of the BEC components is reduced to $N/5$, thus lowering the condensation temperature by 41.5%, which will result in the loss of some of the BEC atoms until the condensate returns to thermal equilibrium.

The optical lattice will slowly drift relative to the BEC and harmonic trap [107], which will cause the interaction and coupling strengths between atoms in the lattice and those in the BEC to fluctuate over time, and between realizations. In our computations, we assumed that, with a proper feedback system, the lattice position could be stabilized to within $0.1\lambda/2$ [85, 107]. The BEC has a TF radius of 18.1 μm $\gg \lambda$. Consequently, we find that the effects of the uncertainty in the lattice position on coupling to the BEC during gate operations should be negligible.

Finally, the background gas of thermal atoms can interact with the BEC and the lattice atoms [54]. The number of thermal atoms and their (non-uniform) density profile will fluctuate over time and between experimental realizations [108, 109]. This will introduce extra noise to the qubit parameters in Eq. (7) due to interactions between the thermal atoms and the atoms in $|0\rangle$, $|1\rangle$ and $|2\rangle$.

### 5.4 Uncertainty in the control fields

The last set of error channels are due to noise in the optical, MW and RF fields that are used to trap and perform operations on our qubits. The first drawback is the necessity of using a spin-dependent optical lattice to realize the model system described in section 2. This introduces several problems that other lattice-based qubits can (mostly) avoid.

The first is that fluctuations in the magnetic field, $\boldsymbol{B}$, will cause the qubit transition frequency to change as well, because the energy difference between $|0\rangle$ and $|1\rangle$ (see Fig. 6) is set by the Zeeman effect [72]. If the magnetic field has an uncertainty of ±4.3 nT [110], then the qubit transition frequency will fluctuate by ±30 Hz. The second problem is that intensity fluctuations in the lattice beams (±312 mW/cm$^2$) will also cause the qubit transition frequency to vary (±28 Hz). In lattice-based qubit platforms that do not require a spin-dependent lattice, the fields and the atomic levels that correspond to the qubit states may be chosen such that the transition frequency will be (to good approximation) intensity and magnetic-field insensitive [111–113].

Any errors in the rotation of the lattice beams' polarization vectors, $\theta$, will cause the amplitude of the lattice potential to change (see Eq. (14)). For atoms in $|1\rangle$, an uncertainty in $\theta$ of ±$\pi/150$ will result in an uncertainty in the qubit transition frequency of ±11.2 Hz (its impact on the Rabi frequency will be negligible). Our choice to use lattice beams with $\lambda = 790$ nm ensures that BEC atoms should not see a lattice potential regardless of $\theta$.

Finally, we discuss errors that arise when addressing individual qubits. We note that these errors arise in all lattice-based qubits, and are no different in our platform than any other. Single-qubit gates in a 3D optical lattice with 5 μm spacing have been realized using a pair of tightly focused optical beams with an average gate fidelity of 0.9962 for target atoms, and an average crosstalk fidelity of 0.9979 [87]. In [85], single sites in a 2D 532 nm lattice were addressed using an optical beam and MW field, which had a 95% gate fidelity and a negligible impact on nontarget atoms. We note that our scheme does not require a 532 nm lattice, and that the qubit atoms could instead be contained in an optical dipole trap array with a larger spacing between qubits to reduce crosstalk [105, 114].

## 6   Conclusion

We have proposed using a 3D spin-dependent optical lattice as a quantum register, and a spatially overlapping BEC in a harmonic trap as a reservoir of atoms. Single-qubit operations

would be realized by loading single atoms from the BEC into individual lattice sites. By coupling the qubits to the BEC or empty modes of the harmonic trap, we showed how CNOT and $\sqrt{\text{SWAP}}$ gates could be implemented between arbitrary pairs of qubits without the qubits ever being brought outside of the two-level qubit basis. We also discussed how nondestructive measurements could be carried out without heating up the atoms, and what the various sources of decoherence in such a system would be.

Compared to Rydberg-based qubits, this platform has slower gate times and computational states that have greater sensitivity to fluctuations in both the magnetic field and the lattice beams. However, our qubits are much better protected against decoherence due to atom losses, they never need to be removed from the ground state manifold, can be measured nondestructively, and a register of over $1,000$ fully-connected qubits is achievable.

The proposed system represents a new pathway for realizing a neutral atom quantum computer, which attempts to address many of the problems currently facing cold atom-based qubits, but there is much research still to be done. Work on addressing individual qubits by coupling them to a non-uniform BEC (thereby removing the need for tightly focused optical beams), and using the BEC as a sensor to probe changing experimental parameters so that computation errors may be corrected in real-time, is already underway. In the future, we hope to experimentally realize this system and demonstrate its potential to help push the field forward.

# 7 Acknowledgments

We would like to thank Maitreyi Jayaseelan, Joseph D. Murphree, and Nathan Lundblad for helpful conversations. This work was supported by NSF grant PHY 1708008, NASA/JPL RSA 1656126.

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
