# Peer review of "A quantum register using collective excitations in a Bose--Einstein condensate"

_SciPost Physics_

## Round 1 · Referee Report · Anonymous (Referee 1) · 2023-1-11

Report

In their manuscript the authors propose a protocol that utilizes collective atomic excitations in order to create single qubit and entangling gates in a cold atom quantum register. An emphasis is made on the point that no Rydberg blockade is needed.

As far as I understand the idea is to create a state-dependent lattice within a BEC and to encode qubits in the individual lattice sites. Entangling gates between qubits are then mediated via the BEC.

I think the idea can be interesting, but in my view the paper is incomprehensible in its current form and cannot be accepted for publication. Here are some examples/reasons explaining why I arrive at this conclusion:

1) The current way in which the paper is written, i.e. the decomposition into small sections and small appendices, does not work. I think the appendices have to be combined with the main text in order to enhance readability.

2) I question the choice of equations that are shown in the manuscript: What do I need Eq. (1) for? Also, I think it would be good to see details of the derivation of Eq. (4), which appears to be the most important one of the paper.

3) I find the notation confusing. In Eq. (5) you define \Omega_si^rj, but actually you are using \Omega_0, \Omega_1, \Omega_2 in the text. I think this can be fine, but I think the more appropriate choice would be to formulate the theory in terms of a minimal model that only contains the states which are really necessary? This would make the whole discussion far more comprehensible and would also avoid rather cumbersome equations such as Eq. (5).

4) Another example for the lack of detail is a statement in Sec. IV: "...for the lattice sites farthest from the center of the trap, is 1/364." As a reader one wonders where this comes from and what the significance of this statement is.

5) A further example is given by the way the magnetic field is discussed: it is introduced in Eq. (1), then it is mentioned in the context of spin-changing collision and later it is mentioned again in the context of the single qubit rotations. When reading the text one actually wonders whether this is always the same magnetic field or not because the story is not told in a coherent way.

I think that this is all fixable. My suggestion is to write the paper in a way in which one introduces the general system at the beginning and then develops an effective few-level model from which the theory should be derived in a comprehensible fashion.

I also have a technical question: How come that in the inequality \Omega_0 << U/hbar (Sec IV) there is no enhancement factor \sqrt{N} appearing?

A final comment: the authors may want to include the following reference: Collectively encoded Rydberg qubit, Physical Review Letters 127, 063604 (2021)

---

## Round 2 · Referee Report · Anonymous (Referee 2) · 2023-5-28

Strengths

This is a novel and interesting scheme for quantum computation with cold trapped atoms.

Weaknesses

The proposed realization of the quantum gates is not sufficiently clear and convincing.

Report

The authors present a new scheme for quantum computation involving cold atoms in an optical lattice potential immersed in a Bose-Einstein condensate. The presence or absence of atoms in individual lattice sites serves as the qubit basis while the BEC plays the role of a reservoir and quantum bus to implement quantum gates.

This is an interesting proposal, but I cannot support the publication of this manuscript in its present form due to numerous shortcomings detailed below.

In the discussion on the realization of the CNOT gate, I am not sure how the second-order coupling between states |n_0,1,n_2> and |n_0-1,1,n_2+1> is obtained and is equal to Ω^2(U_11 - Δ)/(U_11 + Δ)/Δ. This has to be presented in a more clear way. In the Hamiltonian (5) and below, there is no term involving U_11, which I assume should be there to describe the energy of two-atom state in the tight OL potential. Also, the two-atom state in the tight trap should involve additional U_01 and U_12 interactions with the BEC states, which do seem to have been taken into account.

The second part of the CNOT gate is also not clear. Starting with BEC populations 4N/5 and N/5 in states |0> and |2>, why would one expect it to continuously flow to N/5 and 4N/5 if the qubit site is empty? Do the authors assume that U_00 = U_22=U_02 so both components of the BEC remain resonant for any occupations? This does not seem to be the case, based on the parameters of the system listed in Sec. 4.
Finally, how come N is known and can be controlled to such a high precision? If we have many qubits in various states, than the number of atoms in reservoir BEC is not well defined and the gate procedure cannot be implemented. This is only qualitatively mentioned in Sec. 5, while more careful discussion is needed.

I see similar problems with the implementation of the SWAP: The authors assume no atoms in state 2, but the other BEC mode with atoms in state |0> cannon be empty, otherwise how one would create and annihilate atoms in the qubit state |1>. Hence, the interaction of qubit atoms with the atoms in state |0> should be taken into account while the total number of atoms is not a well-defined quantity.
It is not clear to me what is \hat{\Omega} at the end of that section.
Finally, since the SWAP gate involves atom exchange between two distant lattice sites via a common mode, or a set of common modes, one should include retardation coming from the finite velocity of propagation of atoms in space between the two sites, while what is presented in this section reads like the exchange is instantaneous (like photons in the cavity). The SWAP gate times much smaller than the inverse frequency of the HO (the bus mode), as given in sec 4.2 (~100μs), cannot be right: the atom exchanged between the two lattice sites needs at least 2π/ω time to propagate between distant positions in the trap.

Less critical issues:

The last sentence of the abstract is not clear. How can one remove atoms from the computational basis? What is meant by "both"?

In the third sentence of the introduction, by spin state of the atoms the authors perhaps mean the internal state.

On the collective atomic qubits and the problems associated with the uncertainly in the number of atoms, the author may wish to cite npj Quantum Information 6, 54 (2020) and consult PRA 89, 013419 (2014), where this issue was addressed.

By second quantized Hamiltonian, we usually mean the Hamiltonian expressed in terms of creation and annihilation operators for different well defined modes. Eq. (1) is written in terms of the field operators and therefore it is misleading to call it a second quantized Hamiltonian.

In the next paragraph, the spacing between the internal energy levels of the atoms cannot be non-linear; it can be unequal, incommensurate, unharmonic etc.

After Eq. (3), l is not a dimension but direction or axis.

Before noting the bosonic enhancement factor for the atoms in a BEC, the authors should specify that N is the number of such atoms.

After Eq. (4) the indices s and r denote the internal atomic states, while indices i and j denote the lattice sites, while before that equation the notation convention was exactly the opposite (i was the internal state and s the lattice site). This is confusing and should be corrected.

Single site addressability with lasers in an OL with period 500nm is not easy and the finite overlap of the laser with the neighboring lattice sites will be a source of errors.

Requested changes

The realisation of quantum gates should be revised and convincingly presented.
Discussion of various errors should be made more quantitative.
The presentation should be improved.

---

## Round 2 · Referee Report · Anonymous (Referee 3) · 2023-6-8

Strengths

1 - original idea

Weaknesses

1 - clarity in the overall presentation

Report

In their work, the authors propose using BECs to implement a register of qubits. In particular, they show how fundamental operations, such as single qubit rotations, the CNOT gate and the SWAP gate are implemented.
The idea is potentially interesting, but the manuscript lacks clarity. In particular, while a lot of experimental numbers are provided in the last section of the paper, peculiar points regarding the main idea of the proposal are not completely clear to me. Tehrefore, I would suggest publication after the following issues are solved.

Requested changes

-1 The physical system that encodes the qubit is not clear to me. In the abstract, the authors say first that the qubit is represented by an ensemble of atoms and then that a qubit is identified with a lattice site. In the text, the authors focus on a single site, identifying the two states |up> and |down> with the population of a single site with respect to species |1>.

-2 It is not clear to me whether species 0 and 2 play the role of an atom bath or they are strictly associated with the definition of the state of a single qubit. Consequently, in the CNOT gate description, it is first of all unclear how the authors identify the two qubits. Is the second qubit identified with another site of the optical lattice? How do they address a specific site instead of another?

-3 (connected to the previous point) The authors say that their platform can host up to 1000 qubits. Even without a complete analysis of the scalability of the model, describing how two CNOT gates could be independently applied on two couples of qubits would help.

-4 The authors compare their proposal with others based on neutral atoms, presenting as a strength the fact that they don't have the problem of escaping atoms, as it happens instead in the case of other platforms. Can they estimate the probability of having a spurious empty/filled site - errors in single site addressing in optical lattices?

-5 The initialization to the fractions 4/5N and N/5 is not justified.

-6 I would suggest changing the title. Generally, the title anticipates the techniques, the contents, or the main results of a paper, while in this case, the authors mention something (the blockade mechanism in Rydberg atoms) not related to their work.

  • validity: ok
  • significance: ok
  • originality: ok
  • clarity: low
  • formatting: good
  • grammar: good

Author:  Elisha Haber  on 2023-07-12  [id 3798]

(in reply to Report 2 on 2023-06-08)

Reply to referee 2

We gratefully thank the referee for critically reviewing our manuscript, and believe that, after revising it in response to each of their comments, it has been notably improved. Below, for each comment, we provide a point-by-point reply and describe what changes were made in the new manuscript.

-1 The physical system that encodes the qubit is not clear to me. In the abstract, the authors say first that the qubit is represented by an ensemble of atoms and then that a qubit is identified with a lattice site. In the text, the authors focus on a single site, identifying the two states $|\uparrow\rangle$ and $|\downarrow\rangle$ with the population of a single site with respect to species $|1\rangle$.

We thank the referee for alerting us that such a critical part of our proposal was not clear. We have modified the manuscript by including additional details of how the qubit states are defined when they are introduced in section 3.1. We have amended the paragraph following Eq. (7) to include the following sentences:

This choice of qubit states means that single-qubit gates correspond to coherently taking one atom out of the BEC in the HO and placing it in a particular lattice site, or vice versa. Each single-qubit operation leads to every atom in the BEC being placed in the same superposition of occupying or not occupying that particular lattice site. A second single-qubit operation on a different site will produce an even larger superposition in which every atom has some probability of occupying each of the two lattice sites. In this way, the BEC acts as a reservoir of excitations for the array of qubits.

We hope that this clarifies that each lattice site is a qubit, and that the BEC of atoms acts as a reservoir for each of them.

-2 It is not clear to me whether species 0 and 2 play the role of an atom bath or they are strictly associated with the definition of the state of a single qubit. Consequently, in the CNOT gate description, it is first of all unclear how the authors identify the two qubits. Is the second qubit identified with another site of the optical lattice? How do they address a specific site instead of another?

As discussed in response to the first comment, the BEC in $|0\rangle$ and $|2\rangle$ does act as an atom bath, which is connected to every qubit in the $10 \times 10 \times 10$ array. In section 3.2 on the CNOT gate, we have added this sentence to the first paragraph:

Note that the control and target qubits correspond to different sites in the lattice.

We have also added additional details on how different qubits may be addressed. In section 3.1 when we first start discussing qubit gates, we have added this sentence to the last paragraph:

In section 4.2, we also discuss how different sites may be individually addressed in our lattice, and thus how a quantum register with multiple, independently controllable qubits could be realized.

Finally, in section 4.2, we have expanded our discussion of how specific sites are addressed. The second and third paragraphs of the section (as they did before) mention how individual lattice sites could be addressed using either an atom chip, or pairs of tightly focused optical beams, and then give experimental numbers for such beams. In addition, we have added this sentence to the end of the third paragraph:

It is assumed that the addressing beams can be rapidly reconfigured to focus on any qubit in the 3D lattice, for instance, by using a lens on a translation stage and MEMS mirrors as in [86,87,89].

In the fourth to last paragraph of section 4.2 we also added additional details about how individual lattice sites could be addressed specifically during a CNOT gate. Finally, in the third to last paragraph we added a sentence about using pairs of reconfigurable, tightly-focused optical beams to address specific sites during the $\sqrt{\mathrm{SWAP}}$ gate, and in the second paragraph of section 4.3 we added a sentence about addressing sites during the measurement process.

We emphasize that our platform does not require a unique addressing scheme, and that the same methods used in other 3D lattices [1,2,3] would also work in ours.

-3 (connected to the previous point) The authors say that their platform can host up to 1000 qubits. Even without a complete analysis of the scalability of the model, describing how two CNOT gates could be independently applied on two couples of qubits would help.

We thank the referee for pointing out this lack of clarity in our proposal, and refer back to the changes we made to section 4.2 to elucidate the existing schemes for addressing individual sites in an optical lattice, as discussed in response to the second comment above.

-4 The authors compare their proposal with others based on neutral atoms, presenting as a strength the fact that they don't have the problem of escaping atoms, as it happens instead in the case of other platforms. Can they estimate the probability of having a spurious empty/filled site - errors in single site addressing in optical lattices?

In the new manuscript we have expanded the last paragraph of section 5.4 to include a discussion of the single-qubit gate fidelities of existing addressing schemes. In particular, we quote the experimental gate fidelities from [2] (where the authors used a 3D lattice with a 5 $\mu$m spacing) and [4] (where they used a 2D lattice with a $532$ nm spacing). We expect that since the same addressing schemes would work in our platform, similar fidelities could be achieved. In addition, we note at the end of the paragraph that our platform does not require an optical lattice, and that an optical dipole trap array with a larger spacing between qubits could be used instead.

-5 The initialization to the fractions 4/5N and N/5 is not justified.

We thank the reviewer for pointing this out. In the fourth from last paragraph of section 3.2 on the CNOT gate we say:

Assuming all of the population initially begins in either $|0\rangle$ or $|2\rangle$, a significant fraction may initially be transferred directly to the other state using the $\Omega_{02}$ field.

And in the first paragraph after Eq. (10) in the same section we note:

Under the single-mode approximation (SMA) [64], the entire population may be continuously transferred between the two wells/internal states [65].

Finally, in the third paragraph of section 5.3, we include:

The CNOT gate, which is described by a Hamiltonian whose elements are given by Eq. (8), will also be sensitive to fluctuations in $N$. The first step of the CNOT gate is to initialize the BEC with $N_0 = N/5$ and $N_2 = 4N/5$, which cannot be done perfectly given our limited knowledge of $N$. If, as suggested in section 3.2, we apply the field $\Omega_{02} = 1$ kHz to drive population between $|0\rangle$ or $|2\rangle$, then $\sqrt{N}$ fluctuations between realizations will introduce an uncertainty of $2.11 \times 10^{-7}$ in the fraction of atoms in $N_0$ and $N_2$.

Thus, we simulated initializing the population fractions to $N/5$ and $4N/5$, and even with an uncertainty in $N$ of $\sim \sqrt{N}$, the simulation shows that we can expect errors of only $\sim 2.11 \times 10^{-7}$.

-6 I would suggest changing the title. Generally, the title anticipates the techniques, the contents, or the main results of a paper, while in this case, the authors mention something (the blockade mechanism in Rydberg atoms) not related to their work.

We thank the referee for this suggestion, and we have modified the title to be:

A quantum register using collective excitations in a Bose–Einstein condensate

thereby removing any reference to Rydberg atoms.

  1. Y. Wang, X. Zhang, T. A. Corcovilos, A. Kumar and D. S. Weiss, Coherent addressing of individual neutral atoms in a 3D optical lattice, Phys. Rev. Lett. 115, 043003 (2015), doi:10.1103/PhysRevLett.115.043003

  2. Y. Wang, A. Kumar, T.-Y. Wu and D. S. Weiss, Single-qubit gates based on targeted phase shifts in a 3D neutral atom array, Science 352(6293), 1562 (2016), doi:10.1126/science.aaf2581

  3. C. Knoernschild, C. Kim, F. P. Lu and J. Kim, Multiplexed broadband beam steering system utilizing high speed MEMS mirrors, Opt. Express 17(9), 7233 (2009), doi:10.1364/OE.17.007233

  4. C. Weitenberg, M. Endres, J. F. Sherson, M. Cheneau, P. Schauß, T. Fukuhara, I. Bloch and S. Kuhr, Single-spin addressing in an atomic Mott insulator, Nature 471(7338), 319 (2011), doi:10.1038/nature09827

Author:  Elisha Haber  on 2023-07-12  [id 3797]

(in reply to Report 2 on 2023-06-08)

Reply to referee 1

We gratefully thank the referee for critically reviewing our manuscript, and believe that, after revising it in response to each of their comments, it has been notably improved. Below, for each comment, we provide a point-by-point reply and describe what changes were made in the new manuscript.

In the discussion on the realization of the CNOT gate, I am not sure how the second-order coupling between states $|n_0,1,n_2\rangle$ and $|n_0-1,1,n_2+1\rangle$ is obtained and is equal to $\Omega^2 (U_{11} - \Delta) / (U_{11} + \Delta) / \Delta$. This has to be presented in a more clear way.

We thank the referee for suggesting that we clarify this part of the paper. In the new manuscript, section 3.2 has been heavily modified starting from paragraph 4 to better explain how the coupling between $|n_0,1,n_2\rangle$ and $|n_0-1,1,n_2+1\rangle$ was obtained.

In the Hamiltonian (5) and below, there is no term involving $U_{11}$, which I assume should be there to describe the energy of two-atom state in the tight OL potential. Also, the two-atom state in the tight trap should involve additional $U_{01}$ and $U_{12}$ interactions with the BEC states, which do seem to have been taken into account.

The Hamiltonian in Eq. (5) does not contain any $U_{11}$ because, as noted in the paragraph directly above it, we have restricted the dynamics to the subspace with either $0$ or $1$ atom(s) in each lattice site. Eq. (5) therefore only applies to the single-qubit gates in section 3.1, and not to the CNOT gate in section 3.2, where interactions between atoms in the same site must be taken into account. For the small driving fields considered in section 4.2, the approximation of dropping the $U_{11}$ terms (and extra $U_{01}$ and $U_{12}$ terms) from Eq. (5) for single-qubit gates is well justified.

In section 3.2 on the CNOT gate, the $U_{11}$, $U_{01}$ and $U_{12}$ terms are all taken into account in the Hamiltonian given in Eq. (8).

The second part of the CNOT gate is also not clear. Starting with BEC populations $4N/5$ and $N/5$ in states $|0\rangle$ and $|2\rangle$, why would one expect it to continuously flow to $N/5$ and $4N/5$ if the qubit site is empty? Do the authors assume that $U_{00} = U_{22} = U_{02}$ so both components of the BEC remain resonant for any occupations? This does not seem to be the case, based on the parameters of the system listed in Sec. 4.

In response to this question we have clarified the relevant part of section 3.2. We have added a paragraph after Eq. (10) in the new manuscript, which clarifies that we are making the single-mode approximation, in which case the entire population may be transferred between $|0\rangle$ and $|2\rangle$ without requiring $U_{00} = U_{22} = U_{02}$ [1]. Additionally, we have added Fig. 3, which shows the population of $|0\rangle$ during the first step of the CNOT gate as given by (8) in the mean-field limit. Fig. 3 indeed displays complete population transfer (when $\sigma_1 = 0$), despite $U_{00} \neq U_{22} \neq U_{02}$.

The reason such transfer is possible can be seen in the mean-field limit after approximations in Eq. (9) & (10) have been made. In this case, you arrive at the usual double-well Hamiltonian but with these three different interaction energies. After some algebra, one finds that the different $U_{00}$, $U_{22}$, and $U_{02}$ lead to a modified interaction energy and offset between the two wells, the latter of which can be compensated for by introducing a small two-photon detuning to the driving fields, $\Omega_{01}$ and $\Omega_{12}$, thereby allowing the complete population transfer shown in Fig. 3.

Finally, how come $N$ is known and can be controlled to such a high precision? If we have many qubits in various states, than the number of atoms in reservoir BEC is not well defined and the gate procedure cannot be implemented. This is only qualitatively mentioned in Sec. 5, while more careful discussion is needed.

We thank the review for pointing out that this important component of our proposal was not clear in the original manuscript. Throughout the paper, we assume that $N$ is not known to better than $\sqrt{N}$ of the true value, as is typical in BEC experiments. The reason that this uncertainty in $N$ has such a small impact on our qubit gates is because we assume that $N \sim 10^6$ atoms, and thus the relative fluctuations are around $1/\sqrt{N} \sim 0.001$. In paragraphs 2, 3 and 4 of section 5.3 we quantify the error the uncertainty in $N$ will have on each of our quantum gates.

I see similar problems with the implementation of the SWAP: The authors assume no atoms in state 2, but the other BEC mode with atoms in state $|0\rangle$ cannot be empty, otherwise how one would create and annihilate atoms in the qubit state $|1\rangle$. Hence, the interaction of qubit atoms with the atoms in state $|0\rangle$ should be taken into account while the total number of atoms is not a well-defined quantity.

We thank the reviewer for pointing out that we did not take the interactions between the BEC atoms in $|0\rangle$ and the qubit atoms into account in our previous manuscript. We failed to do so because these interactions have a marginal impact on the Hamiltonian in Eq. (12) that describes this process. In the new manuscript, these interactions are explicitly accounted for in Eq. (11), and in Eq. (12) they are absorbed into $\Delta$, $\Delta'$ and $\omega_n$, showing that their inclusion results in (small) modifications to the single-photon detunings, and the HO energies. In the last three paragraphs of section 4.2, the numbers and calculations for the $\sqrt{\mathrm{SWAP}}$ gate now take these interactions into account.

It is not clear to me what is $\hat{\Omega}$ at the end of that section.

In the last paragraph of section 3.3 we have clarified that $\hat{\Omega}$ is the spatial profile of the field, and that it was originally defined in Eq. (8).

Finally, since the SWAP gate involves atom exchange between two distant lattice sites via a common mode, or a set of common modes, one should include retardation coming from the finite velocity of propagation of atoms in space between the two sites, while what is presented in this section reads like the exchange is instantaneous (like photons in the cavity). The SWAP gate times much smaller than the inverse frequency of the HO (the bus mode), as given in sec 4.2 ($\sim 100$ $\mu$s), cannot be right: the atom exchanged between the two lattice sites needs at least $2\pi/\omega$ time to propagate between distant positions in the trap.

We thank the reviewer for drawing our attention to this oversight. In our original manuscript, we did not clearly describe how we are able to transfer atoms between lattice sites so quickly. We can do this because the HO levels, $\omega_{\mathrm{n}}$ in Eq. (12), are not harmonic during the gate operation. If they were, then, as the referee has pointed out, we could not drive the atom between sites faster than $2\pi/\omega$. In [2], this is shown to be a consequence of the single-photon detuning, $\Delta$ (and $\Delta'$ in our case), being much larger than the energy level spacing. This limitation is explained, and [2] is cited, in our new manuscript in the second to last paragraph of section 3.3.

In the last paragraph of 3.3 we describe how applying an off-resonant $\Omega_{02}$ field to connect the BEC in $|0\rangle$ and the one or zero atom(s) in $|2\rangle$ causes the HO levels, $\omega_{\mathrm{n}}$, to shift. The level spacing becomes anharmonic, and many of the levels become separated by hundreds of kHz or more, which allows us to drive this two-photon transition strongly enough to have a gate time (for certain pairs of qubits) on the order of $100$ $\mu$s. We note that, except for the extra coupling to the BEC, our approach is identical to that of [2].

Less critical issues:

The last sentence of the abstract is not clear. How can one remove atoms from the computational basis? What is meant by "both"?

We thank the reviewer for pointing out that this part of the abstract wasn't clear. We have modified this sentence to read:

In this setup, the effect of atom losses has been mitigated, the atoms never need to be removed from the ground state manifold, and separate storage and computational bases for the qubits are not required, all of which can be significant sources of decoherence in many other types of atomic qubits.

Which we hope that this clarifies our message. In particular, that our qubits never need to be brought outside of the qubit basis, which distinguishes this platform from, for example, many types of Rydberg qubits. In Rydberg-based approaches, two hyperfine levels are usually used to encode the qubit states, and the atoms need to be brought outside of this basis and into a highly excited Rydberg level during two-qubit operations (see e.g. [3]).

In the third sentence of the introduction, by spin state of the atoms the authors perhaps mean the internal state.

We thank the reviewer for pointing out this error, and we have corrected it in the new manuscript.

On the collective atomic qubits and the problems associated with the uncertainty in the number of atoms, the author may wish to cite npj Quantum Information 6, 54 (2020) and consult PRA 89, 013419 (2014), where this issue was addressed.

We thank the review for suggesting we add these references, and we have included them as references [105] and [106].

By second quantized Hamiltonian, we usually mean the Hamiltonian expressed in terms of creation and annihilation operators for different well defined modes. Eq. (1) is written in terms of the field operators and therefore it is misleading to call it a second quantized Hamiltonian.

In the next paragraph, the spacing between the internal energy levels of the atoms cannot be non-linear; it can be unequal, incommensurate, unharmonic etc.

After Eq. (3), l is not a dimension but direction or axis.

Before noting the bosonic enhancement factor for the atoms in a BEC, the authors should specify that $N$ is the number of such atoms.

After Eq. (4) the indices s and r denote the internal atomic states, while indices i and j denote the lattice sites, while before that equation the notation convention was exactly the opposite (i was the internal state and s the lattice site). This is confusing and should be corrected.

We have corrected each of these mistakes in the new manuscript.

Single site addressability with lasers in an OL with period 500nm is not easy and the finite overlap of the laser with the neighboring lattice sites will be a source of errors.

We thank the reviewer for pointing this out, and have included additional discussion of it in the last paragraph of section 5.4. In this paragraph, we provide the experimental gate fidelities for addressing atoms in similar optical lattices to ours, and at the end we note that the optical lattice is not a necessary component of our protocol and could be replaced by an optical tweezer trap array with a larger spacing.

  1. J. Williams, R. Walser, J. Cooper, E. Cornell and M. Holland, Nonlinear Josephson-type oscillations of a driven, two-component Bose--Einstein condensate, Phys. Rev. A 59, R31936 (1999), doi:10.1103/PhysRevA.59.R31.93728.

  2. A. B. Deb, G. Smirne, R. M. Godun and C. J. Foot, A method of state-selective transfer of atoms between microtraps based on the Franck–Condon principle, Journal of Physics B: Atomic, Molecular and Optical Physics 40(21), 4131 (2007), doi:10.1088/0953-4075/40/21/001.

  3. M. Saffman, Quantum computing with atomic qubits and Rydberg interactions: progress and challenges, Journal of Physics B: Atomic, Molecular and Optical Physics 49(20), 202001 (2016), doi:10.1088/0953-4075/49/20/202001.

---

## Round 2 · Author Response

Dear Editor,

Thank you for soliciting reports from referees, and for recommending that we revise our manuscript. We are also very grateful to the referee for providing constructive comments on our submission. We have revised our manuscript in response to each of the referee's comments, and believe that it has been significantly improved.

We have provided a short summary of the changes we made below, and a more detailed list in our reply to the referee report.

Gratefully,
Elisha Haber, Zekai Chen, and Nicholas P. Bigelow

---

## Round 2 · List of Changes

In response to referee comment (1), we integrated all of the appendices in the old manuscript into the body of the new one (or removed the content of the appendix entirely). We also edited the manuscript to enhance its overall readability.

In response to comment (2), we reorganized the discussion of the model we used and provided additional details of its derivation.

In response to comment (3), we now derive and discuss a minimal model that can be used to understand the more complicated and realistic system that we consider later.

In response to comment (4), we clarified the particular sentence in the paper, and expanded our discussion of it.

In response to comment (5), when discussing the applied magnetic field, we always denote it by its variable to remove any ambiguity.

We also fixed the typo pointed out by the referee in one of our inequalities, and included their suggested reference.

Finally, in addition to the changes suggested by the referee, we also made several additional edits, which are described at the bottom of our reply to the referee report.

---

## Round 3 · Referee Report · Anonymous (Referee 4) · 2023-8-7

Strengths

This is a novel and interesting scheme for quantum computation with cold trapped atoms.

Weaknesses

The manuscript should be further improved to become more clear and convincing.

Report

The authors have addressed much of the criticism in my previous report and the revised manuscript is much more clear. Yet, some issues still remain and the paper should be further improved to be acceptable for publication, as detailed below.

In the first sentence of the abstract, ensemble qubits based on Rydberg blockade are mentioned, but it is not so relevant to the present study to mention it in the most prominent place.

The HO potential for atoms in the lattice will induce energy shift which should be taken into account. The parameter ε in Eq. (4) is in fact that energy which is neglected in the rest of the paper.

Equation (4) (and (11)) is not exactly Bose-Hubbard Hamiltonian (the hopping term between the lattice sites is missing, while the driving field Ω acts locally on the atoms and does not induce intersite tuneling). But it is second-quantized Hamiltonian.

For the number of atoms in states 1,2,3 the authors sometimes use lowercase n, sometimes uppercase N, which is confusing. Then, in section 5, lowercase n is used for qubit number.

The explanation of the CNOT gate is more clear now but still confusing. In the text the authors explain that for the qubit state |spin-down>, after the transfer, the BEC in state 0 will have population N/5 and the BEC is state 2 population 4N/5. Now the control qubit is driven from the BEC 0 or BEC 2? I assume 2, as shown in Fig. 4 third column, but the text in the last paragraph of this section says the opposite. The authors should correct that and also correct the qubit states in that paragraph.

I am still not convinced that the SWAP gate will function as argued by the authors. First, using largely detuned laser coupling to the dense spectrum of many HO levels does not work, because these levels have alternating parity and the sum in the first line of Eq. (12) becomes very small due to the partial cancellation of the different transition paths, as the authors also mention. On the other hand, the brief description of how one can make it work in the last paragraph of section 3.3 is not clear and probably not correct. Even if the laser detuning becomes smaller than the level separation to break the destructive interference of the many paths, the retardation for the atom travelling between the two distant lattice sites will be much longer than the 82.7μs. A simple physics tells me that for the atom to travel between two sites separated by some distance d in time t, it should have the large velocity v=d/t or kinetic energy mv^2/2 = recoil energy h^k^/2m that it can only obtain by absorbing a photon with wavevector k from the laser or MW field Ω, which is small.
I suggest the authors remove this section and all the discussion of the SWAP gate from the paper.

The detunings Δ_1,2 in Eq. (13) and in Eqs. (14-15) are not the same as they correspond to different lasers creating the harmonic trap and lattice potential. This should be clarified.

In Sec. 4.2, the authors discuss the implementation of single qubit gates with global RF+MW fields (making only one selected site resonant using focused laser beams). But they also mention Raman transition with two laser beams, for which the described formalism involving (N)^0.5 Ω enhancement of the Rabi frequency does not apply, since the laser does not irradiate the whole BEC of N atoms. I assume the Raman transition is mentioned only to contrast with the RF+MF approach and the authors do not mean to use it.

Requested changes

Make corrections and clarifications requested in the report.

Remove sec. 3.3 and all the discussion of the SWAP gate from the paper.

---

## Round 3 · Referee Report · Anonymous (Referee 5) · 2023-9-19

Report

The authors have addressed the points raised in the previous report, therefore I suggest publication.

---

## Round 3 · Author Response

Dear Editor,

Thank you for soliciting reports from referees, and for recommending that we revise our manuscript. We are also grateful to the referees for providing constructive comments on our submission, we have revised our manuscript accordingly, and believe that it has been significantly improved.

We have provided a short summary of the changes we made below, and a more detailed list in our replies to the referee reports.

Gratefully,
Elisha Haber, Zekai Chen, and Nicholas P. Bigelow

---

## Round 3 · List of Changes

We thank referee 1 for their recommended changes, and have revised our manuscript in the following ways:

(1) Section 3.2 on the CNOT gate has been heavily modified to be more clear.

(2) Section 5.3 has been expanded to include quantitative error estimates that arise from the sqrt(N) uncertainty in N.

(3) Sections 3.3 and 4.2 have been modified to take into account the interactions between the BEC in state |0> and atoms in states |1> and |2> during the sqrt{SWAP} gate.

(4) The end of section 3.3 was changed to make it more clear what hat{Sigma} is and how atoms may be transferred between lattice sites so quickly during the sqrt{SWAP} gate.

(5) The last sentence of the abstract has been changed.

(6) The word 'spin' in the third paragraph of the introduction has been changed to 'internal.'

(7) The papers the reviewer suggested have been cited.

(8) We have removed the reference to Eq. (1) as a second quantized Hamiltonian right before Eq. (1).

(9) We have changed the description of the internal energy level spacing of the atoms to 'anharmonic' in section 2.

(10) The l variable after Eq. (3) is now defined to be an axis.

(11) N is now defined in the same sentence that the sqrt{N} Bosonic enhancement factor is mentioned in section 2.

(12) The confusing indices in Eq. (4) have been changed to be consistent with what came before.

(13) Additional discussion on single site addressing and gate fidelities has been added to the last paragraph of section 5.4.

We thank referee 2 for their suggested changes, and have revised our manuscript in the following ways:

(1) Additional details about how the qubit states are defined are given in section 3.1, following Eq. (7).

(2) & (3) In the first paragraph of section 3.2 we added a sentence clarifying that the control and target qubits of the CNOT gate correspond to two different lattice sites. Additional details about addressing schemes during each of the different qubit gates considered in this paper were added to section 4.2.

(4) We have added discussion on gate fidelities in the last paragraph of section 5.4.

(5) We justified initializing specific fractions of the BEC population in states |0> and |2> during the CNOT gate by including additional discussion at the end of section 3.2. We also simulated this process in the mean-field limit and gave the estimated error in the initialization fraction in section 5.3.

(6) We removed the reference to the Rydberg blockade from the title of the new manuscript.

---

## Editorial Decision

resubmitted